



# GOCO06s – A satellite-only global gravity field model

Andreas Kvas[1], Jan Martin Brockmann[2], Sandro Krauss[1,5], Till Schubert[2], Thomas Gruber[3],
Ulrich Meyer[4], Torsten Mayer-Gürr[1], Wolf-Dieter Schuh[2], Adrian Jäggi[4], and Roland Pail[3]

[1]Institute of Geodesy, Graz University of Technology, Steyrergasse 30/III, 8010 Graz, Austria
[2]Institute of Geodesy and Geoinformation, University of Bonn, Nußallee 17, 53115 Bonn, Germany
[3]Astronomical and Physical Geodesy, Technical University of Munich, Arcisstraße 21, 80333 Munich, Germany
[4]Astronomical Institute, University of Bern, Sidlerstrasse 5, 3012 Bern, Switzerland
[5]Austrian Academy of Sciences, Space Research Institute, Schmiedlstraße 6, 8042 Graz, Austria

**Correspondence:** Andreas Kvas (kvas@tugraz.at)

**Abstract.** GOCO06s is the latest satellite-only global gravity field model computed by the GOCO ("Gravity Observation Combination") project. It is based on over a billion observations acquired over 15 years from 19 satellites with different complementary observation principles. This combination of different measurement techniques is key in providing consistent high-accuracy and best possible spatial resolution of the Earth's gravity field.

The motivation for the new release was in the availability of reprocessed observation data for GRACE and GOCE, updated background models, and substantial improvements in the processing chains of the individual contributions. Due to the long observation period, the model consists not only of a static gravity field, but comprises additionally modeled temporal variations. These are represented by time variable spherical harmonic coefficients, using a deterministic model for a regularized trend and annual oscillation.

The main focus within the GOCO combination process is on the proper handling of the stochastic behavior of the input data. Appropriate noise modelling for the used observations result in realistic accuracy information for the derived gravity field solution. This accuracy information, represented by the full variance-covariance matrix, is extremely useful for further combination with, for example, terrestrial gravity data and is published together with the solution.

The primary model data (Kvas et al., 2019b) consisting of potential coefficients representing Earth's static gravity field, together with secular and annual variations are available on International Centre for Global Earth Models (http://icgem.gfz-potsdam.de/, last accessed 2020-06-11). This data set is identified with the DOI 10.5880/ICGEM.2019.002.

Supplementary material consisting of the full variance-covariance matrix of the static potential coefficients and estimated co-seismic mass changes are available on ifg.tugraz.at/GOCO (last accessed 2020-06-11).

## 1 Introduction

Global models of Earth's static gravity field are crucial for geophysical and geodetic applications. These include oceanography (e.g. Knudsen et al., 2011; Rio et al., 2014; Johannessen et al., 2014; Bingham et al., 2014), tectonics (e.g. Johannessen et al., 2003; Ebbing et al., 2018), or establishing global reference and height systems (e.g. Rummel, 2013; Gerlach and Rummel, 2013). A special class of global gravity field models are the models derived solely from satellite observations. In contrast





to terrestrial observations – which are generally sparse and collected by various instruments in different quality levels and
samplings – satellite observations typically cover the entire Earth's surface. As the measurements are captured with the same
sensor platform, the global accuracy level of the observations is much more consistent compared to terrestrial observations.
This also alleviates adequate stochastic modelling at observation level (Ellmer, 2018; Schubert et al., 2019) to subsequently
provide uncertainty information for the derived gravity field.

The era of dedicated gravity field satellite missions (Rummel et al., 2002) provided a huge set of observations of the static as
well as time-variable gravity field. The satellite orbits are sensitive to the long wavelength of the gravity field (e.g. Montenbruck
and Gill, 2000; Rummel et al., 2002). They are either determined from ground by Satellite Laser Ranging (SLR) or from
space via the tracking of a low-earth orbiter (LEO) with the satellites of the Global Position System (GPS) constellation. The
laser tracking of high-flying SLR satellites is sensitive only for the longest wavelengths, but allows to observe the temporal
variations as well (e.g. Maier et al., 2012; Sośnica et al., 2015; Bloßfeld et al., 2018). Starting with the dedicated CHAllenging
Minisatellite Payload satellite mission (CHAMP, Reigbar et al., 1999) different LEO satellites were tracked by GPS. This
technique, called atellite-to-satellite tracking in high-low (SST-hl) configuration, has since been used to derive static, but also
temporal gravity field models of the Earth (e.g. Bezděk et al., 2016; Lück et al., 2018; Teixeira da Encarnação et al., 2016).

Although, SST-hl is sensitive to shorter wavelengths due to the generally lower satellite altitude compared to SLR, the spatial
resolution is still limited. With the Gravity Recovery And Climate Experiment (GRACE) twin satellite mission launched in
2002 (Tapley et al., 2004, 2019), inline satellite-to-satellite tracking (SST-low-low, SST-ll) was established. This then novel
measurement principle relies on very precise measurements of inter-satellite distance variations between a leading and a trailing
satellite in the same orbit. While GRACE is dedicated to observe temporal changes in Earth's gravitational field, typically in
the form of monthly snapshots (e.g. Rummel et al., 2002; Tapley et al., 2004; Mayer-Gürr, 2006; Chen et al., 2018a; Beutler
et al., 2010), accumulating observations over a longer time span allows to derive a mean (static) gravity field. After the end of
the mission in 2017, the GRACE Follow-On mission continues the observations since 2018 (Kornfeld et al., 2019; Landerer
et al., 2020), following the same design principle.

To increase the spatial resolution, tailored to the requirements of geodetic, geophysical and oceanographic applications,
the Gravity field and steady-state Ocean Circulation Explorer (GOCE) mission (Battrick, 1999) was realized and launched in
2009. In addition to SST-hl, a gradiometer served for the first time as a core instrument to measure the second derivatives of
the Earth's gravitational potential in gradiometer reference frame (Rummel and Colombo, 1985; Rummel et al., 2002, 2011b;
Johannessen et al., 2003). Due to the design of the instrument (Stummer et al., 2012; Siemes et al., 2012, 2019), only four out
of the six derivatives of the Marussi tensor could be measured with high-precision ($V_{XX}$, $V_{XZ}$, $V_{YY}$ and $V_{ZZ}$). Due to the
measurement of the second derivatives, the spatial resolution can be increased at the cost of more complex noise characteristics.
Since the determination of the long-wavelength signal is not accurate enough for time-variable gravity inversion, the focus of
GOCE is on the static gravity field. In the same fashion as for the other missions, multiple expert groups perform gravity field
recovery with different processing strategies (e.g. Pail et al., 2011; Farahani et al., 2013; Migliaccio et al., 2011; Brockmann
et al., 2014; Bruinsma et al., 2014; Schall et al., 2014; Yi, 2012).





Within typical processing workflows, mission (or technique) specific gravity field models are derived by technique-specific experts and published as mission/technique-only gravity field models. These models are published in terms of a spherical harmonic expansion and reflect the state-of-the-art analysis. Nowadays, these models are more often equipped with a full covariance matrix, which reflects the error structure specific to the analysis and observation technique (e.g. Brockmann et al., 2014; Bruinsma et al., 2014; Mayer-Gürr et al., 2018a; Kvas et al., 2019a).

With a full covariance matrix in combination with a solution vector, the original system of normal equations can be reconstructed. Together with additional meta-information, such as the number of observations and the weighted square sum of the observation vector, multiple gravity field solutions can be used to derive *combined* gravity field models. As long as the used observations are statistically independent, the solution is the same as when starting from the raw observations. Combining multiple observation principles and satellite missions (LEOs, GRACE, GOCE and SLR) has the advantage that possible weaknesses of the individual contributions can be compensated. However, for a proper relative weighting, it is essential that the used individual models provide a realistic error description in form of the covariance matrix of the spherical harmonic coefficients. In the case of satellite-only gravity field models, which are currently given to at most degree 300, SLR typically dominates the very long wavelengths (degree 2 to 5), the medium wavelengths are mainly determined by GRACE (up to degree 130), and GOCE is the main contributor to the medium to short wavelengths (degree 130 to 300) (e.g. Pail et al., 2011; Maier et al., 2012; Yi et al., 2013).

Primary focus of combined models is the static gravity field (e.g. Pail et al., 2010; Farahani et al., 2013; Yi et al., 2013), nevertheless some of the more recent models also incorporate temporal variations, for example (piecewise) linear trends or annual oscillations (e.g. Rudenko et al., 2014). Since 2011, various models and approaches combining mainly GRACE and GOCE were published. Whereas Farahani et al. (2013) did a joint least squares analysis of GRACE and GOCE observations spending a lot of effort on the stochastic modelling of the observations, Yi et al. (2013) combined their self-computed GOCE normal equations with the publicly available ITG-Grace2010 model.

A specific and updated series of combined satellite-only models is provided by the EIGEN-*S series which started as a pure CHAMP only model (Reigber et al., 2003) and turned to a GRACE, GOCE and SLR combination model in the latest EIGEN-6S release (Förste et al., 2016). These models are closely related to the GOCE solutions determined with the direct approach, which are already combination models (Bruinsma et al., 2014, 2013). Basically, GRACE normal equations processed by GFZ (Dahle et al., 2019) are combined with GOCE normal equations assembled with the so called direct approach (Bruinsma et al., 2014; Pail et al., 2011) and SLR normal equations. The combination is performed on normal equation level. To counteract the non-perfect stochastic modelling in the GRACE, GOCE and SLR analysis, the combination is performed only for certain manually chosen degree ranges, which are selected according to the strengths of the individual techniques.

This paper continues the global satellite only models of the GOCO*s series, which are produced by the Gravity Observation COmbination (GOCO) Consortium (GOCO, 2017). Since 2010, a series of satellite only models (Pail et al., 2010) were computed by the group, which is composed by experts on SST-hl, GRACE, GOCE and SLR based gravity field determination. The GOCO models combine the GRACE models produced by Graz University of Technology (previously ITG Bonn, now ITSG-series, Mayer-Gürr et al. (2018b); Kvas et al. (2019a)) with the time-wise GOCE normal equations (Brockmann et al.,





2014), SST-hl normal equations of several LEOs (Zehentner and Mayer-Gürr, 2016) and since the second release GOCO02S SLR normal equations (Maier et al., 2012). Within the assembly of all the used individual models, a lot of effort is spend on the data-adaptive stochastic modelling of the observation error characteristics. Consequently, all input models come along with a realistic covariance matrix and are thus well suited as input for the applied combination procedure.

GOCO satellite-only models are widely used and accepted as state of the art by the community. The models published so far (GOCO01s, GOCO02s, GOCO03s, and GOCO05s) were used in a wide range of applications, for example, for global and local high resolution models (e.g. Huang and Véronneau, 2013; Pail et al., 2016, 2018; Klees et al., 2018; Slobbe et al., 2019), geophysical studies (e.g. Rummel et al., 2011a; Abrehdary et al., 2017; Chen et al., 2018b), oceanographic studies (e.g. Farrell et al., 2012; Siegismund, 2013) or geodetic height unification (e.g. Vergos et al., 2018).

Within this contribution, the latest release – GOCO06S (Kvas et al., 2019b) – is presented. For that purpose, the data used to estimate the global satellite-only model are discussed in Sect. 2. Sect. 3 covers the parametrization, which was used to describe the gravity field as well as its temporal changes. Furthermore the applied methods are discussed. The properties of the final estimate are discussed in Sect. 4 and validated with state of the art concepts. Sect. 5 summarizes the main characteristics of the model and presents some conclusions following from the analysis.

## 2 Data sources and background models

### 2.1 Gravity field and steady-state Ocean Explorer (GOCE)

The GOCO models make use of the normal equations of the time-wise gravity field models (e.g. Pail et al., 2011; Brockmann et al., 2014). They are independent from any other gravity field information and only contain measurements collected by GOCE. Furthermore, due to the use of an advanced stochastic model for the gravity gradients during the assembly of the SGG normal equations, the formal errors are a realistic description of the uncertainties which is beneficial for the combination procedure.

Within GOCO06s, normal equations from the time-wise RL06 model (GO_CONS_EGM_TIM_RL06, Brockmann et al., 2019) are used, which are based on satellite gravity gradient (SGG) observations of the latest reprocessing campaign in 2019. A single unconstrained normal equation assembled from the $V_{XX}$, $V_{XZ}$, $V_{YY}$ and $V_{ZZ}$ gravity gradient observations up to spherical harmonic degree 300 enters the computation of GOCO06s (cf. Sect. 3).

Due to the sensitivity of the gravity gradients and their spectral noise characteristics, normal equations are assembled only with respect to the static part of the Earth's gravity field. To define a clear reference epoch and being consistent to the GRACE processing, time-variable gravity signal was reduced from the along track gravity gradient observations in advance. For this purpose, the models as summarized in Tab. 1 were used to refer the observations (and thus the right hand side $\mathbf{n}_{\text{SGG}}$ to the reference epoch 2010-01-01. It enters the combination procedure (cf. Sect. 3) as a normal equation system of the form

$$\mathbf{N}_{\text{GOCE}}\hat{\mathbf{x}}_{\text{GOCE}} = \mathbf{n}_{\text{GOCE}} \tag{1}$$



**Table 1.** Used models to reduce the temporal gravity changes.

| Force | Model | Reference |
|---|---|---|
| Earth's gravity field (annual oscillation & trend) | GOCO05s | Mayer-Gürr et al. (2015) |
| Non-tidal atmosphere and ocean variations & atmospheric tides | AOD1B RL06 | Dobslaw et al. (2017) |
| Astronomical tides (Moon, Sun, planets) | JPL DE421 | Folkner et al. (2009) |
| Solid earth tides | IERS2010 | Petit and Luzum (2010) |
| Ocean tides | FES2014b | Carrere et al. (2015) |
| Pole tides | IERS2010, linear mean pole | Petit and Luzum (2010) |
| Ocean pole tides | Desai (2002), linear mean pole | Desai (2002) |

where $\hat{x}_{\text{GOCE}}$ is the vector of unknown static spherical harmonic coefficients, $\mathbf{N}_{\text{GOCE}}$ and $\mathbf{n}_{\text{GOCE}}$ are the normal equation coefficient matrix and right-hand-side respectively. This system of normal equations is given in a predefined ordering for the GOCE contribution (spherical harmonic degrees 2 to 300). This single normal equation already contains the stochastic models used in the time-wise processing (cf. Schubert et al., 2019), the relative weighting of the different gravity gradients and the different time-periods of observations. It is the same normal equation as used in GO_CONS_EGM_TIM_RL06, which is in the context of a GOCE-only model combined with SST-hl data and constrained.

## 2.2 Gravity Recovery and Climate Experiment (GRACE)

The GRACE contribution to GOCO06s consists of a subset of the normal equations of ITSG-Grace2018s (Mayer-Gürr et al., 2018; Kvas et al., 2019a). Similarly to the GOCE normal equations, ITSG-Grace2018s features a full stochastic model of the used inter-satellite range-rates, hence provides realistic formal errors. The used normal equations incorporate GRACE data in the time span of April 2002 to August 2016, because at the time of the computation, the reprocessed observation data (, JPL) for the single-accelerometer months starting from 2016-09 to 2017-06 were not yet publicly available. The normal equations feature a static part parametrized up to degree/order 200 and secular and annual variations up to degree/order 120. The background models used in the processing of the GRACE data are consistent with the GOCE contribution (see Kvas et al., 2019a, for details). The normal equations

$$\mathbf{N}_{\text{GRACE}}\hat{x}_{\text{GRACE}} = \mathbf{n}_{\text{GRACE}} \tag{2}$$

are computed in monthly batches and then accumulated over the observation time span. Due to the high sensitivity of GRACE to the longer wavelengths of the spherical harmonic spectrum, it is the primary contributor to the temporal variations.





### 2.3 Kinematic orbit of Low Earth orbit (LEO) satellites

For GOCO06s, we made use of kinematic orbit positions from 9 satellites including CHAMP, GRACE A/B, TerraSAR-X, TanDEM-X, GOCE, and SWARM A+B+C. The kinematic orbit positions were computed using precise point positioning following the approach of Zehentner and Mayer-Gürr (2016). The normal equations $\mathbf{N}_k^{(s)}$, $\mathbf{n}_k^{(s)}$ for gravity field determination were assembled using the short-arc approach (Mayer-Gürr, 2006; Teixeira da Encarnação et al., 2020), in monthly batches for each satellite $s$ and epoch $k$. For CHAMP, GRACE, TerraSAR-X, TanDEM-X and SWARM we set up the normal equations up to degree and order 120 while for GOCE, due to the lower orbit altitude and the subsequent higher sensitivity, the normal equations were assembled up to degree and order 150. For all LEO satellites, only the static gravity field was modeled. Since we applied an appropriate stochastic model in the assembly of the systems of normal equations, the relative weighting between the satellites and epochs is already contained in the normal equations. Therefore, we only need to accumulate all satellites and months, as

$$\mathbf{N}_{\mathrm{LEO}} = \sum_k \sum_s \mathbf{N}_k^{(s)}, \qquad \mathbf{n}_{\mathrm{LEO}} = \sum_k \sum_s \mathbf{n}_k^{(s)}. \tag{3}$$

The kinematic orbits provide gravity field information primarily to the lower (near-) sectorial coefficients of the spherical harmonic spectrum, to which the GRACE inter-satellite range-rates are less sensitive.

### 2.4 Satellite Laser Ranging (SLR)

To stabilize the long wavelength part of the spectrum, we added SLR observations to the combination. The used observations match the GRACE time span from 2002-04 to 2016-08 and feature a total of 10 satellites, including LAGEOS 1/2, Ajisai, Stella, Starlette, LARES, LARETS, Etalon 1/2 and BLITS. The SLR observations were processed in weekly batches consisting of three 7-day arcs and one arc of variable length to complete the month, resulting in systems of normal equations $\mathbf{N}_k^{(s)} \hat{\mathbf{x}}_k^{(s)} = \mathbf{n}_k^{(s)}$ up to d/o 60 for each satellite $s$. To obtain the normal equations for static, trend, and annual oscillation from the weekly systems of normal equations we first perform a parameter transformation. We express the weekly potential coefficients $\mathbf{x}_k^{(s)}$ in terms of static, trend and annual oscillation as

$$\mathbf{x}_k^{(s)} = \mathbf{x}_{\mathrm{static}}^{(s)} + \mathbf{x}_{\mathrm{trend}}^{(s)}(t_k - t_0) + \mathbf{x}_{\mathrm{cos}}^{(s)} \cos(\omega(t_k - t_0)) + \mathbf{x}_{\mathrm{sin}}^{(s)} \sin(\omega(t_k - t_0)), \tag{4}$$

or equivalently in matrix notation

$$\mathbf{x}_k^{(s)} = \underbrace{\left[ \mathbf{I} \quad \mathbf{I} \cdot (t_k - t_0) \quad \mathbf{I} \cdot \cos(\omega(t_k - t_0)) \quad \mathbf{I} \cdot \sin(\omega(t_k - t_0)) \right]}_{:=\mathbf{F}} \begin{bmatrix} \mathbf{x}_{\mathrm{static}}^{(s)} \\ \mathbf{x}_{\mathrm{trend}}^{(s)} \\ \mathbf{x}_{\mathrm{cos}}^{(s)} \\ \mathbf{x}_{\mathrm{sin}}^{(s)} \end{bmatrix}. \tag{5}$$

Here, $\mathbf{I}$ is an identity matrix of appropriate dimension, $t_0$ is the reference epoch of the combined gravity field and $\omega$ is the angular frequency corresponding to an oscillation of 365.25 days. Using this parameter substitution we can transform the



weekly systems of normal equations and accumulate over whole time span,

$$\mathbf{N}_{\mathrm{SLR}}^{(s)} = \sum_k \mathbf{F}^T \mathbf{N}_k^{(s)} \mathbf{F}, \qquad \mathbf{n}_{\mathrm{SLR}}^{(s)} = \sum_k \mathbf{F}^T \mathbf{n}_k^{(s)}. \tag{6}$$

After the accumulation we determined the relative weight between the individual SLR satellites $s$ by variance component estimation (VCE). Since we processed all satellites consistently, it is reasonable to assume that the formal errors are underestimated in the same fashion for all SLR targets, even though no proper stochastic model for the SLR observation was used in the processing. To determine the relative weights $w^{(s)}$ of the satellites, we do not make use of the full system of normal equations, but only use the static part up to degree and order 10. Additionally, we applied a Kaula constraint to stabilize the system of equations. The final normal equation matrix and right-hand-side used in the combination procedure is then computed as the weighted sum of all satellites,

$$\mathbf{N}_{\mathrm{SLR}} = \sum_s w^{(s)} \mathbf{N}^{(s)}, \qquad \mathbf{n}_{\mathrm{SLR}} = \sum_s w^{(s)} \mathbf{n}^{(s)}. \tag{7}$$

## 3  Combination process

We combined the individual contributions from GOCE, GRACE, kinematic LEO orbits, SLR, and constraints on the basis of normal equations using VCE. VCE is a widely used technique in geodesy for combining different observation groups (Koch and Kusche, 2002; Lemoine et al., 2013; Meyer et al., 2019). The general idea is to determine the relative weights $w_*$ between heterogeneous observation types within the same least squares adjustment where the unknown parameters, in our case the gravity field, are estimated. Typically, this is an iterative procedure where initial weights are refined. The inverse of these weights, called variance factors, are the quotient of a residual square sum and the redundancy of the observation group. In the most general form this quotient is given as

$$\sigma_k^2 = \frac{\hat{\mathbf{e}}^T \mathbf{\Sigma}^{-1} \mathbf{V}_k \mathbf{\Sigma}^{-1} \hat{\mathbf{e}}}{\mathrm{trace}\left\{ (\mathbf{\Sigma}^{-1} - \mathbf{\Sigma}^{-1} \mathbf{A}_k \bar{\mathbf{N}}^{-1} \mathbf{A}_k^T \mathbf{\Sigma}^{-1}) \mathbf{V}_k \right\}}, \tag{8}$$

where $\hat{\mathbf{e}}$ are post-fit residuals, $\mathbf{\Sigma} = \sum_k \sigma_k^2 \mathbf{V}_k$ is the compound covariance matrix and $\mathbf{A}_k$ is the design matrix of the $k$-th observation group. The iterative nature of VCE can be seen in Eq. (8) where the compound covariance matrix computed from the variance factors of the previous iteration is necessary to compute the new values. For observation groups which are given as normal equations, so all satellite contributions and also the Kaula constraint, this simplifies to

$$\sigma_k^2 = \frac{(\mathbf{l}^T \mathbf{P} \mathbf{l})_k - 2\mathbf{n}^T \hat{\mathbf{x}} + \hat{\mathbf{x}}^T \mathbf{N}_k \hat{\mathbf{x}}}{n - \mathrm{trace}\left\{ \bar{\mathbf{N}}^{-1} \mathbf{N}_k \right\} \cdot \sigma_k^{-2}}. \tag{9}$$

The general form presented in Eq. (8) comes into play when we determine the weights of the regionally varying constraints for the trend and annual signal (see Section 3.1). There we find that, for example, the variance factors for the trend constraint can be computed from

$$\sigma_k^2 = \frac{\hat{\mathbf{x}}_{\mathrm{trend}}^T \mathbf{\Sigma}_\Omega^{-1} \mathbf{V}_k \mathbf{\Sigma}_\Omega^{-1} \hat{\mathbf{x}}_{\mathrm{trend}}}{\mathrm{trace}\left\{ [\mathbf{\Sigma}_\Omega^{-1} - \mathbf{\Sigma}_\Omega^{-1} (\bar{\mathbf{N}}_{\mathrm{trend}} + \mathbf{\Sigma}_\Omega^{-1})^{-1} \mathbf{\Sigma}_\Omega^{-1}] \mathbf{V}_k \right\}}. \tag{10}$$





Here, $\hat{\mathbf{x}}_{\mathrm{trend}}$ are the estimated trend parameters and $\bar{\mathbf{N}}_{\mathrm{trend}}$ is the accumulated system of normal equations of all satellites. The determination of the relative weights $w_* = \frac{1}{\sigma_*^2}$ is a key criteria for the overall solution quality. The GOCE, GRACE and kinematic orbit contributions feature a proper stochastic model of the used observables and therefore realistic formal errors.

This is a prerequisite for a proper determination of relative weights using VCE (Meyer et al., 2019). The SLR contribution used here lacks an adequate stochastic observation model which results in formal accuracies which are too optimistic. To counteract the high weights arising from this mismodeling, the SLR system of normal equations was artificially downweighted by a factor of 10 in each iteration step. The full, accumulated system of normal equations solved in each iteration step is given by

$$\bar{\mathbf{N}} = w_{\mathrm{GOCE}}\mathbf{N}_{\mathrm{GOCE}} + w_{\mathrm{GRACE}}\mathbf{N}_{\mathrm{GRACE}} + w_{\mathrm{LEO}}\mathbf{N}_{\mathrm{LEO}} + w_{\mathrm{SLR}}\mathbf{N}_{\mathrm{SLR}} + w_{\mathrm{Kaula}}\mathbf{K} + \boldsymbol{\Sigma}_{\mathrm{trend}}^{-1}(\mathbf{w}_{\mathrm{trend}}) + \boldsymbol{\Sigma}_{\mathrm{annual}}^{-1}(\mathbf{w}_{\mathrm{annual}}), \qquad (11)$$

$$\bar{\mathbf{n}} = w_{\mathrm{GOCE}}\mathbf{n}_{\mathrm{GOCE}} + w_{\mathrm{GRACE}}\mathbf{n}_{\mathrm{GRACE}} + w_{\mathrm{LEO}}\mathbf{n}_{\mathrm{LEO}} + w_{\mathrm{SLR}}\mathbf{n}_{\mathrm{SLR}}. \qquad (12)$$

It features all satellite contributions as well as constraints for certain parameters. In order to reduce the power in the higher spherical harmonic degrees and specifically the polar regions where the GOCE gradiometer observation provide little to no information, the solution is zero-constrained using a Kaula-type signal model for degrees higher than 150. This Kaula regularization is represented by the matrix $\mathbf{K}$. Furthermore, the co-estimated trend and annual oscillation are also zero-constrained

with a regionally varying regularization matrix $\boldsymbol{\Sigma}_{\mathrm{trend}}^{-1}$ and $\boldsymbol{\Sigma}_{\mathrm{annual}}^{-1}$ respectively. The vectors $\mathbf{w}_{\mathrm{trend}}$ and $\mathbf{w}_{\mathrm{annual}}$ contain the weights for each modeled region (see Section 3.1).

The accumulation of the individual contributions in Eq. (11) assumes that the individual normal equations are of the same dimension. Since the maximum expansion degree differs between the techniques and not all contribute to the temporal variations, the systems of normal equations have to be zero-padded. The structure of the combined normal equation coefficient matrix is

depicted in Figure 1. We estimate 211 788 parameters in total with 90 597 to describe the static gravity field and 121 191 parameters for the co-estimated temporal variations. Another 121 191 parameters representing co-seismic mass change caused by major earthquakes (see section 3.3) have been preeliminated. The upper triangle of the (symmetric) normal equation coefficient matrix therefore requires 167 GB of memory. The published system of normal equations only features the unconstrained static part with all temporal variations eliminated and requires only 31 GB.

## 3.1    Regularization of co-estimated temporal variations

In order to properly decorrelate the co-estimated temporal variations from the long-term mean field, we constrained the corresponding parameters. While the previous solution GOCO05s, featured a simple Kaula constraint, GOCO06s employs regionally varying prior information to account for the greatly different signal levels in individual regions. Using a globally uniform signal model, which a Kaula-type regularization provides, damps the secular signal in, for example, Greenland, while overestimating

the expected signal level in the ocean. To avoid this undesired behavior, but still introduce as little prior information as possible, we developed a tailored regularization strategy.

As a first step, the globe was subdivided into regions with similar temporal behavior such as the ocean, Greenland, Antarctica, the Caspian Sea, and the remaining land masses. For each region $\Omega_i$, a signal covariance matrix was derived by applying a window matrix $\mathbf{W}_{\Omega_i}$ to a Kaula-type signal covariance model $\mathbf{K}$. We construct the window matrix in space domain by making





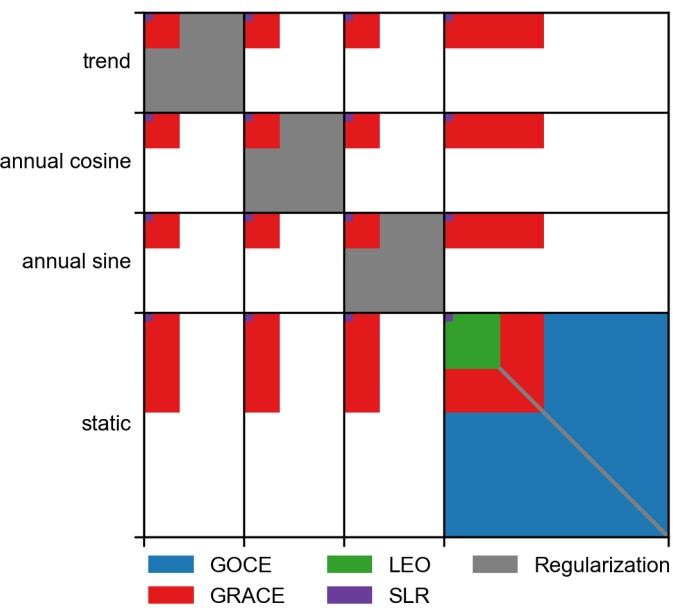

**Figure 1.** Structure of the combined normal equation coefficient matrix, with all contributions overlaid. The different parameter groups for trend, annual, and static potential coefficients are highlighted and plotted to scale.

use of spherical harmonic synthesis and analysis. The general idea is to create an operator which propagates a vector of potential coefficients to source masses on a grid (Wahr et al., 1998), applying a window function to this grid and then using quadrature (Sneeuw, 1994) to transform the windowed source masses back to potential coefficients. All these operations are linear, so we can express the window matrix as a matrix product,

$$\mathbf{W}_{\Omega_i} = \mathbf{A}\mathbf{M}_{\Omega_i}\mathbf{S}. \tag{13}$$

The diagonal matrix $\mathbf{M}_{\Omega_i}$ is a binary window function convolved with a Gaussian kernel featuring a half-width of $220\,\mathrm{km}$ (Jekeli, 1981) to mitigate ringing effects on region boundaries.

Even though the observation contribution to the temporal variations is band-limited to degree and order 120, as outlined in section 2.2, the higher expansion degree of the signal covariance matrix does provide additional information. Specifically, the transition between regions can be modeled sharper thus enabling a better spatial separation. A similar form of this approach,

which combines band-limited observation information with high resolution prior information can be found in Save et al. (2016). The resulting global, compound covariance matrix

$$\mathbf{\Sigma}_\Omega = \sum_i \sigma_i^2 \mathbf{W}_{\Omega_i}\mathbf{K}\mathbf{W}_{\Omega_i}^T = \sum_i \sigma_i^2 \mathbf{V}_i, \tag{14}$$



was then used to formulate a zero-constraint for trend parameters $\mathbf{y}$, with

$$\mathbf{0} = \mathbf{I}\mathbf{y} + \mathbf{v}, \qquad \mathbf{v} \sim \mathcal{N}(\mathbf{0}, \mathbf{\Sigma}_\Omega), \tag{15}$$

where $\mathbf{I}$ is a unit matrix of appropriate dimension. The formulation of the compound covariance matrix in Eq. (14) implies that the individual regions are treated as uncorrelated. This has the side-effect that $\mathbf{\Sigma}_\Omega$ will be dense, even though we use a Kaula-type signal covariance model.

For each region the unknown signal level $\sigma_i$ was determined through VCE during the adjustment process. This means that apart from the isotropic signal model shape, only the geographic location of the region boundaries is introduced as prior
information. The same strategy was used to regularize the annual oscillation, however the globe was subdivided into only two regions, namely continents and the ocean. Furthermore, to conserve the phase of the oscillation, a combined variance factor was estimated for both sine and cosine coefficients. The impact of this novel approach compared to the Kaula-type regularization of the preceding GOCO satellite-only model on the estimated secular variations can be seen in figure 2. It is evident that the noise over the ocean is greatly reduced in GOCO06s compared to its predecessor. In regions where we have a large gradient in the
signal level such as Greenland or the Antarctic Peninsula we observe a better confinement of the signal within the land masses. This means that with the new regularization strategy, less signal leaks from land into the ocean. But there are also limitations to the employed approach. Looking at Greenland, a uniform signal level for the region is not sufficient given the dramatic mass loss at the coasts. Still, there is a clear improvement visible in the new release GOCO06s.

## 3.2 Contribution of individual components

When dealing with combined satellite models, the contribution of each component is of particular interest. It clearly reveals the strengths and weaknesses of the different techniques in determining specific parts of the spherical harmonic spectrum.

We compute the contribution of each component $i \in \{\text{GOCE}, \text{GRACE}, \text{LEO}, \text{SLR}\}$ and the regularization by first assembling the contribution or redundancy matrix

$$\mathbf{R}_i = \bar{\mathbf{N}}^{-1}\mathbf{N}_i. \tag{16}$$

In Eq. (16), $\bar{\mathbf{N}}$ is the combined normal equation coefficient matrix and $\mathbf{N}_i$ is the normal equation coefficient matrix of the $i$-th component.

The main diagonal of $\mathbf{R}_i$ then gives an indication of how much each estimated parameter is informed by the respective component $i$. Since we only look at estimated potential coefficients, it is convenient to depict the contribution to each coefficient per degree and order in the coefficient triangle. Figure 3 shows the contribution to the static gravity field of all contributing
sources. We can clearly see the complementary nature of GRACE and GOCE, where the former is the primary contributor to the long wavelength part of the spectrum up to degree 130. GRACE also constrains and compensates the polar gap of GOCE which can be seen in the higher contribution to the near-zonal coefficients up to degree 200. The GOCE SGG observations dominate degrees 130 to 280. Between degrees 280 to 300, where we approach a signal-to-noise ratio of 1, the effect of regularization starts to play a role. LEO orbits primarily contribute to coefficients up to degree 10 and near-sectorial coefficients up to the

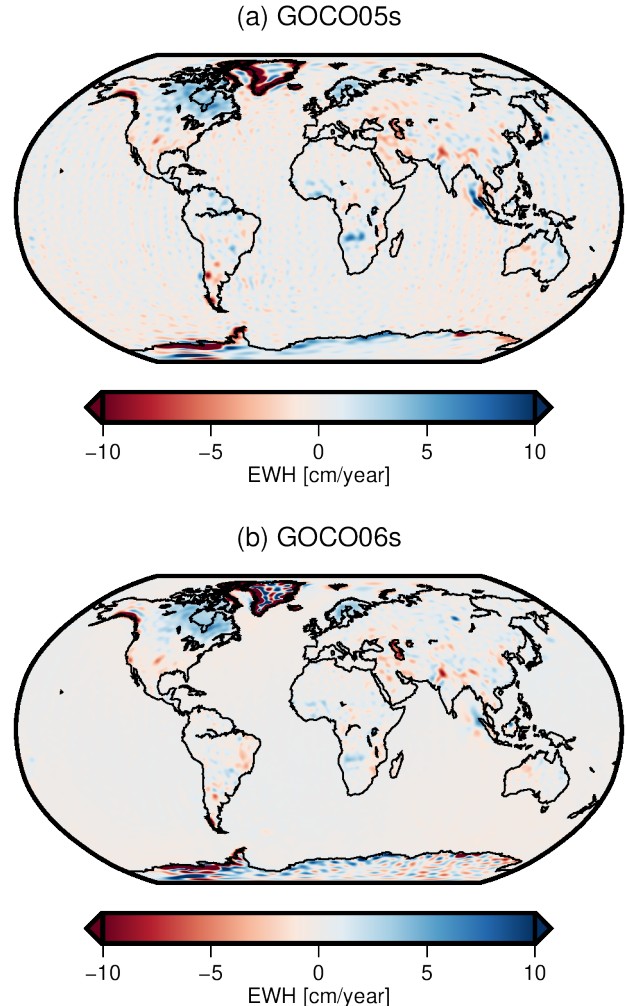

**Figure 2.** Co-estimated secular variation of GOCO05s (a) and GOCO06s (b) in equivalent water heights (EWH).

maximum degree of the involved system of normal equations. The resonance orders of GRACE, which occur on multiples of 15 (Cheng and Ries, 2017), are clearly visible in the contribution plots. These parts of the spectrum are determined less accurately by the inter-satellite ranging data of GRACE (Seo et al., 2008) and are therefore compensated by the other techniques (see e.g. the higher contribution of GOCE in this orders).

     SLR primarily contributes to degree 2 and even zonal coefficients up to degree 12, with a minor contribution to the near-
sectorials around degree 15. These findings are consistent with Bloßfeld et al. (2015).

     We find a similar picture when looking at the contributions for the estimated trend and annual oscillation shown in Figures 4 and 5.



**Figure 3.** Individual contribution to the static part of the estimated potential coefficients for a) GRACE, b) GOCE SGG, c) Kaula regularization, d) LEO orbits, and e) SLR. Note the different axis limits for panels d) and e).

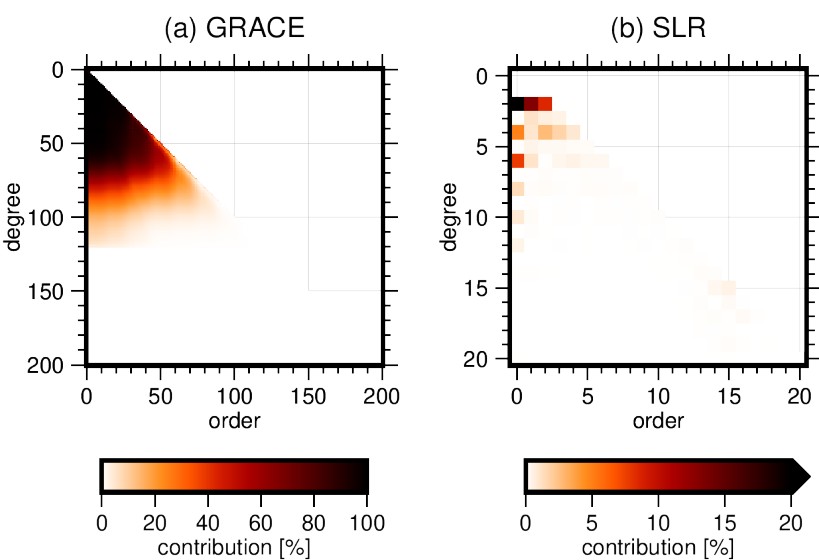

**Figure 4.** Contribution of a) GRACE and b) SLR to the estimated trend. Note the different axis limits.

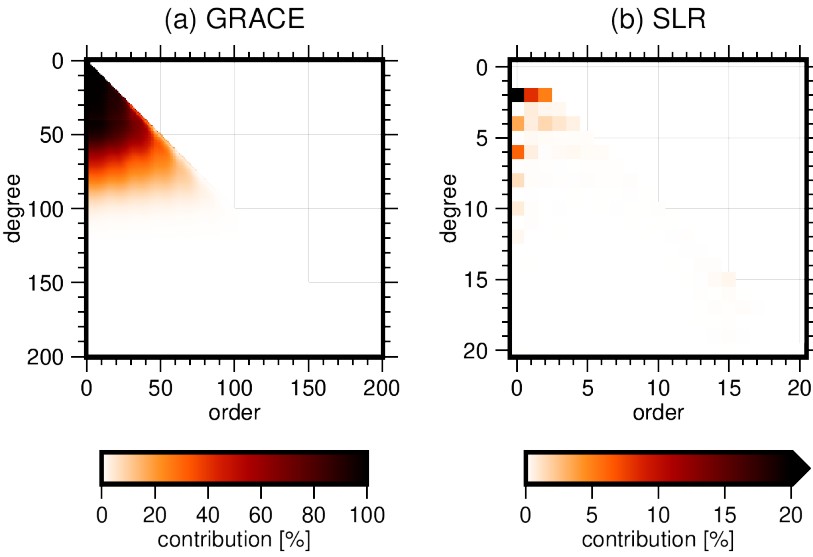

**Figure 5.** Contribution of a) GRACE and b) SLR to the estimated annual oscillation. Note the different axis limits.



### 3.3 Estimation of co-seismic gravity changes

The simple parametrization of Earth's gravity field with static, trend, and annual signal basis functions cannot capture instantaneous gravity changes caused by, for example, large earthquakes. This mismodelling results in an apparent secular variation in the affected regions as the co-seismic gravity change is mapped into the trend estimate. To avoid this behavior, we estimate an additional step function in regions where co-seismic mass change is expected, thus improving the description of the temporal evolution of Earth's gravity field. The methodology is exemplified on the basis of a single earthquake dividing the whole observation time span into two intervals $i = \{1, 2\}$, where interval $i = 1$ refers to observations before the event and $i = 2$ to the observations captured after the event, respectively. But it can be generalized to any number of intervals in a straightforward manner. For each interval we assemble the observation equations

$$\mathbf{l}_i = \mathbf{A}_i \mathbf{x}_i + \mathbf{e}_i \qquad \mathbf{e}_i \sim \mathcal{N}(\mathbf{0}, \mathbf{\Sigma}_i) \tag{17}$$

with $\mathbf{l}_i$ being the observation vector, $\mathbf{A}_i$ the design matrix, $\mathbf{x}_i$ the static gravity field parameters, and $\mathbf{e}_i$ the residual vector. We then form the blocked system of observation equations for the whole observation time span

$$\begin{bmatrix} \mathbf{l}_1 \\ \mathbf{l}_2 \end{bmatrix} = \begin{bmatrix} \mathbf{A}_1 & \\ & \mathbf{A}_2 \end{bmatrix} \begin{bmatrix} \mathbf{x}_1 \\ \mathbf{x}_2 \end{bmatrix} + \begin{bmatrix} \mathbf{e}_1 \\ \mathbf{e}_2 \end{bmatrix}. \tag{18}$$

The next step is to perform the parameter transformation

$$\begin{bmatrix} \mathbf{x}_1 \\ \mathbf{x}_2 \end{bmatrix} = \begin{bmatrix} \mathbf{I} & \mathbf{I} \\ & \mathbf{I} \end{bmatrix} \begin{bmatrix} \mathbf{z} \\ \mathbf{x} \end{bmatrix}, \tag{19}$$

where $\mathbf{I}$ is an identity matrix of appropriate dimension, $\mathbf{x}$ is the static gravity field for the whole time span, and $\mathbf{z}$ is the correction for interval $i = 1$. Substituting Eq. (19) into Eq. (18) yields

$$\begin{bmatrix} \mathbf{l}_1 \\ \mathbf{l}_2 \end{bmatrix} = \begin{bmatrix} \mathbf{A}_1 & \mathbf{A}_1 \\ & \mathbf{A}_2 \end{bmatrix} \begin{bmatrix} \mathbf{z} \\ \mathbf{x} \end{bmatrix} + \begin{bmatrix} \mathbf{e}_1 \\ \mathbf{e}_2 \end{bmatrix}. \tag{20}$$

Since both $\mathbf{x}$ and $\mathbf{z}$ are global representations of Earth's gravity field, the functional model in Eq. (20) would result in a loss of redundancy in regions where no co-seismic change occurred. To counteract this overparametrization, we introduce the pseudo-observations

$$\mathbf{0} = \mathbf{W}\mathbf{z} + \mathbf{w} \qquad \mathbf{w} \sim \mathcal{N}(\mathbf{0}, \mathbf{\Sigma}_{\mathbf{w}}), \tag{21}$$

where $\mathbf{W}$ is a window matrix covering Earth's surface except for the region where a co-seismic change is expected. After combining the pseudo-observations with the transformed observation equations in Eq. (20), the resulting system of normal equations has the structure

$$\begin{bmatrix} \mathbf{N}_1 + \mathbf{W}^T \mathbf{\Sigma}_{\mathbf{w}}^{-1} \mathbf{W} & \mathbf{N}_1 \\ \mathbf{N}_1 & \mathbf{N}_1 + \mathbf{N}_2 \end{bmatrix} \begin{bmatrix} \hat{\mathbf{z}} \\ \hat{\mathbf{x}} \end{bmatrix} = \begin{bmatrix} \mathbf{n}_1 \\ \mathbf{n}_1 + \mathbf{n}_2 \end{bmatrix}. \tag{22}$$



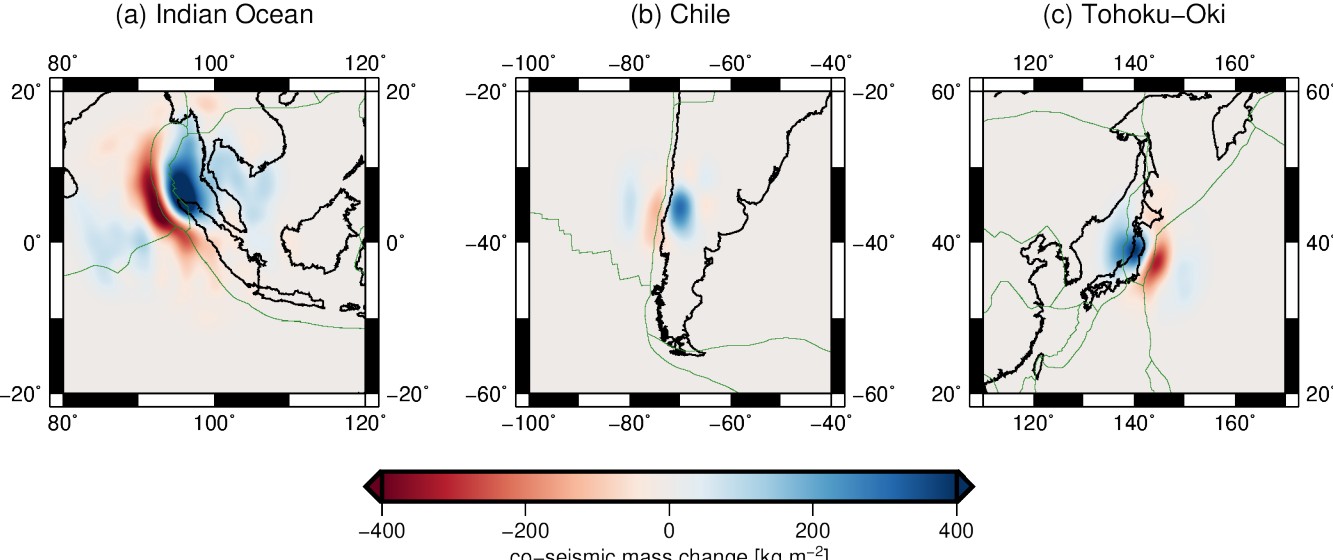

**Figure 6.** Estimated co-seismic mass change for all modelled earthquakes (230 km Gaussian filter applied).

Increasing the weight of the constraint, that is setting $\Sigma_{\mathbf{w}} \to 0$, which in practice is done by scaling with a number close to
310  numerically zero, then allows a signal in $\hat{\mathbf{z}}$ only in regions within the predefined area. This retains the redundancy in points
which are not affected by the earthquake, thus not influencing the estimate $\hat{\mathbf{x}}$ there. We applied this approach to model three
major earthquakes during the GRACE lifespan, specifically, the 2004 Indian Ocean earthquake (Han et al., 2006), the 2010
Chile earthquake (Han et al., 2010), and the 2011 Tohoku-Oki earthquake (Panet et al., 2018). Since GRACE is the primary
contributor to the estimated temporal variations at these spatial scales, the parameter transformation was considered only to
315  the corresponding GRACE normal equations. The estimated co-seismic mass changes can be seen in Figure 6. We can clearly
observe that the estimated signal is spatially confined, therefore, we retain the redundancy outside of the defined area. The
advantages and limitations regarding the estimation of the trend are best exemplified by comparison with GRACE monthly
solution and the previous GOCO05s release (not accounting for co-seismic changes). Figure 7 shows the estimated trends for
GOCO05s and GOCO06s (including the new co-estimated step) together with the time series of monthly solutions from ITSG-
320  Grace2018 (Kvas et al., 2019a), evaluated close to the epicenter of the 2004 Indian Ocean earthquake, where we observe a
large co-seismic mass change. We can clearly observe that adding the co-estimation of co-seismic events greatly improves the
accuracy of the secular variations. However, the monthly solutions show different rates before and after the event, which can
obviously not be modelled by just a uniform trend over the whole observation time span. This is however a deliberate trade-off
to retain redundancy in the trend estimates and simple usability of the dataset.

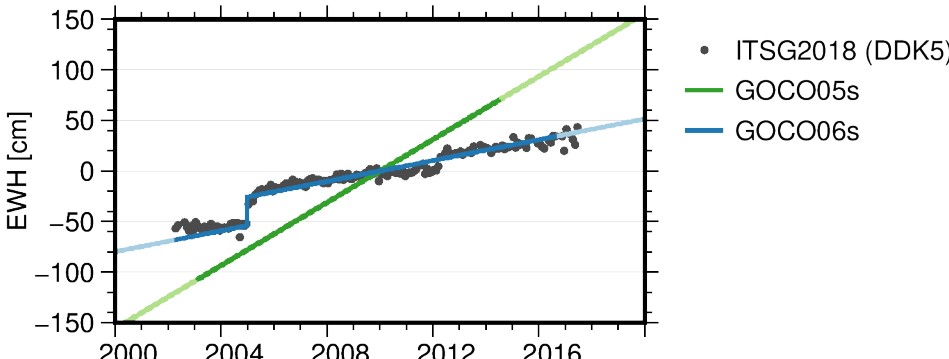

**Figure 7.** Comparison of estimated secular variation from GOCO05s, GOCO06s (including estimated co-seismic mass change) and filtered GRACE monthly solutions in terms of equivalent water height (EWH).

## 4 Results and evaluation

The complete published data set of GOCO06s consists of a static gravity field solution up to degree and order 300, an unconstrained system of normal equations of the static part for further combination, secular and annual gravity field variations up to degree and order 200, and co-seismic mass changes for 3 major earthquakes. All components are published in widely used data formats such as the ICGEM format for potential coefficients (Barthelmes and Förste, 2011) and the SINEX format for normal equations (IERS, 2006).

The data product of primary interest for the community certainly is the estimated static gravity field together with its uncertainty information represented by the system of normal equations. Therefore, we focus our evaluations and discussions on these components. Figure 8 depicts degree amplitudes of differences of state-of-the-art satellite only gravity field models and XGM2016, a gravity field model which combines GOCO05s and terrestrial data (Pail et al., 2016). This is the explanation for the small differences between XGM2016 and GOCO05s in the low degrees where the terrestrial data does not significantly contribute. The other large differences between the compared models in degrees below 60 are primarily explained by the respective reference epochs (for example, 2010-01-01 for GOCO06s and 2010-09-01 for GOCE DIR6). This means, the employed measurement techniques are sensitive to temporal variations in Earth's gravity field, therefore we see signal rather than noise in this frequency band. Concerning the GOCE gradiometer reprocessing we can see improvements from degree 150, where these data start to dominate the solutions. Here, we can clearly distinguish between models based on the new data (GOCO06s, TIM6, DIR6) and models based on the previous release (GOCO05s, EIGEN-GRGS RL04). Figure 9 shows the degree amplitudes of component-wise differences with respect to the combined solution of each component of GOCO06s. We excluded SLR here because the very ill-posed system of normal equations can only be solved up to degree 5-6 without additional information (Cheng et al., 2011). It nicely summarizes the major contributions to the final GOCO06s solution and shows the consistency of formal and empirical errors. It further highlights the importance of stochastic modeling which enables this

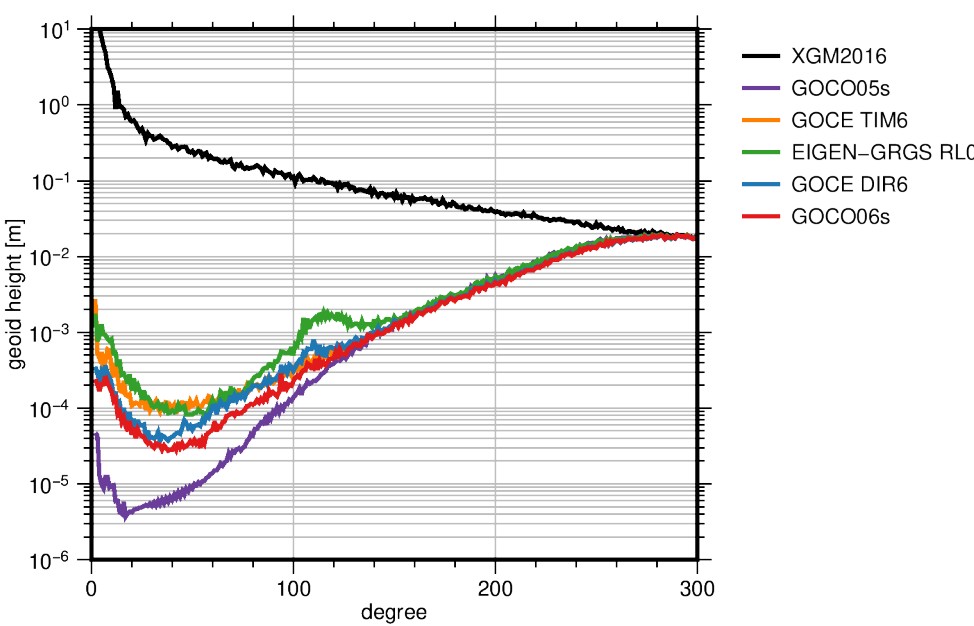

**Figure 8.** Degree amplitudes of various state-of-the-art satellite-only models. (8 degree polar cap excluded).

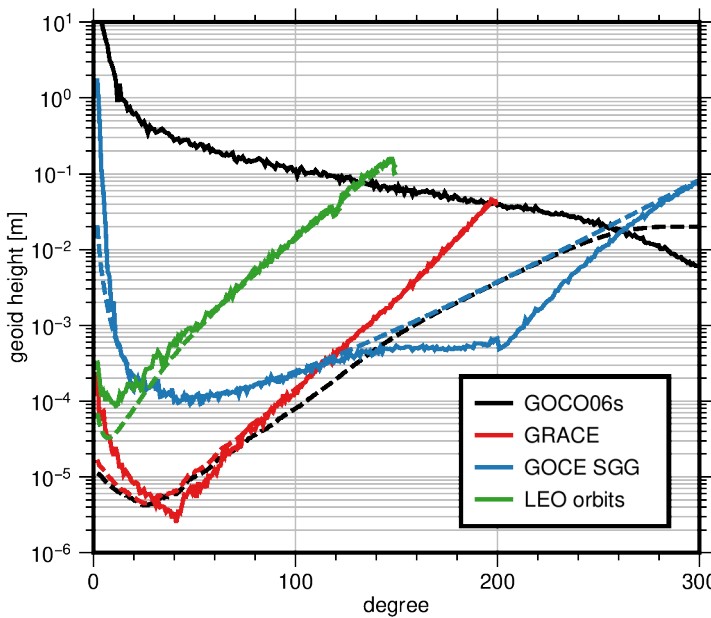

**Figure 9.** Degree amplitudes and corresponding formal errors of the individual GOCO06s components (8 degree polar cap excluded). SLR is not shown because no standalone solution can be computed due to the ill-posed adjustment problem.





consistency. The deviation of the combined solution from the GOCE SGG contribution starting at degree 240 is explained by the Kaula constraint applied to the combination.

## 4.1 Comparison with GNSS leveling

Over land geoid heights (height anomalies or geoid undulations) computed from global gravity models can be validated against
independent geoid observations as they are determined by on ground GNSS levelling. By subtracting physical heights determined by spirit levelling (orthometric or normal heights) from ellipsoidal heights as they are observed with GNSS one can compute such independent geoid heights. The challenge when comparing geoid heights derived from satellite-only gravity field models to observed geoid heights mainly is to eliminate the so-called omission error. This error, which represents the higher resolution geoid signal not observable by satellites due to the gravitational signal attenuation at satellite altitude, causes
the major part of the differences leading to unrealistic quality estimates for the satellite-only gravity model. Therefore, the omission error needs to be estimated from other sources and taken into account prior to computing the differences. For this purpose, high resolution gravity field models as EGM2008 (Pavlis et al., 2012) or XGM2019e (Zingerle et al., 2019, 2020) and an ultra-high resolution model of the gravity signal caused by the Earth's topography as ERTM2160 (Hirt et al., 2014) are used. The omission error computed from these models is subtracted from the observed geoid heights before they are compared to the
satellite-only model geoid heights. Apart from this, a second systematic error might be present in physical heights observed by spirit levelling. Depending on the spirit levelling technique and the carefulness of the observer errors accumulate along the levelling lines leading to artificial tilts (or other systematics) of the physical heights with respect to the reference equipotential surface, which is defined by the height system reference point in a country. Therefore, prior to analyze differences between observed and global model derived geoid heights a correction surface for the geoid differences shall be estimated and applied.
This ensures that errors due to such tilts and due to offsets of regional height systems with respect to the global geoid are eliminated beforehand. The computational procedure to compute and analyze geoid height differences between observed geoid heights and those determined from a global gravity field model is described in detail in (Gruber et al., 2011) and (Gruber and Willberg, 2019).

For validating the GOCO06s model a number of global gravity field models and a number of GNSS levelling data sets is
applied. For gravity field models on one hand the previous satellite-only models of the GOCO series are used (GOCO01s, GOCO02s, GOCO03s and GOCO05s). These models are characterized by combining GRACE and GOCE satellite data with increasing amount of observations and some additional satellite data (cf. Sect. 1). In addition, an independent GRACE/GOCE satellite-only combination model namely GOCE-DIR6 is used for comparisons, which to a large extent is based on the same amount of GRACE and GOCE data as GOCO06s (Förste et al., 2019). In order to identify the performance of pure GRACE
and pure GOCE models compared to the combined GOCO06s model the ITSG-Grace2018s (Kvas et al., 2019a) model and the GOCE-TIM6 (Brockmann et al., 2019) models are applied to the validation procedure. Finally, a model combining the GOCO06s satellite data with surface, airborne and altimetric gravity data is used in order to identify the signal content higher degrees of the GOCO06s model. This is the XGM2019e high resolution global gravity field model (Zingerle et al., 2019, 2020). GNSS levelling data are available from many sources for many areas in the world. Typically, these data sets show



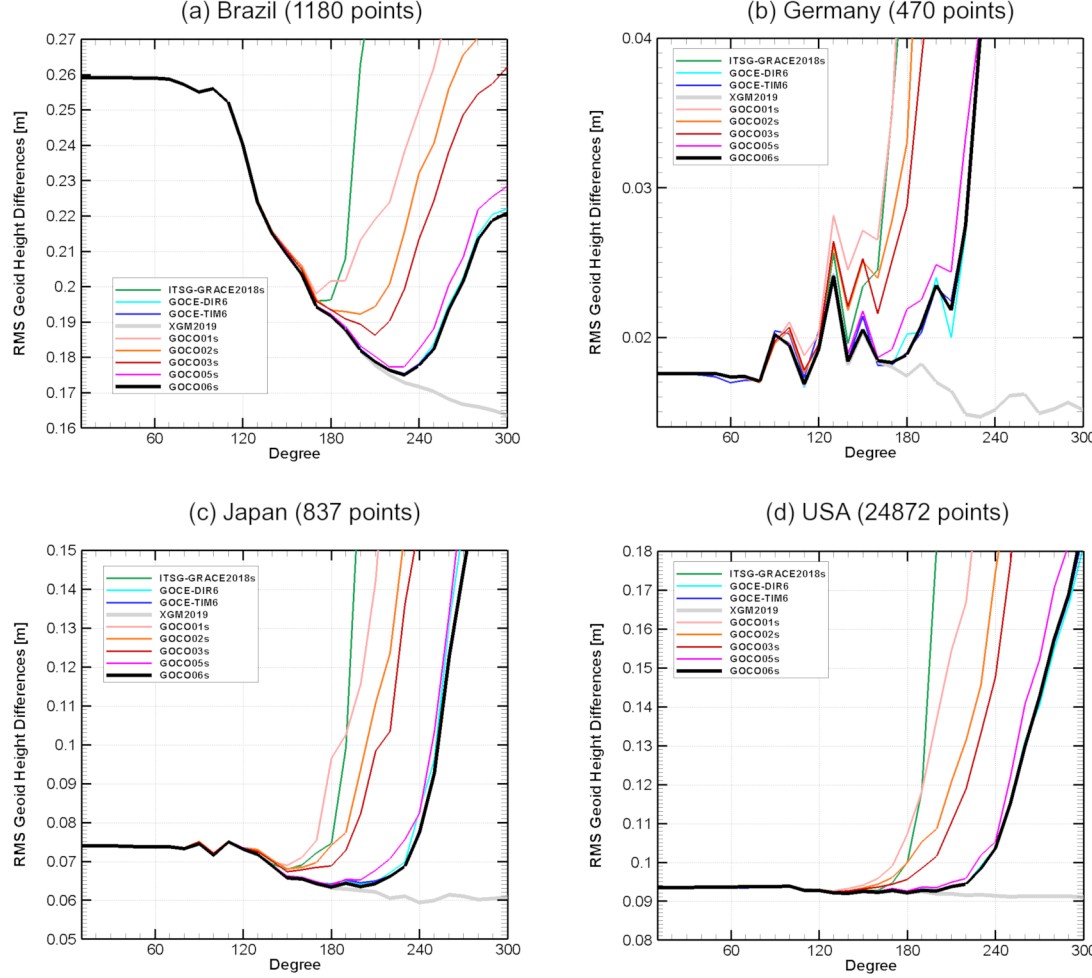

**Figure 10.** RMS of geoid height differences between global gravity models and independently observed geoid heights from GNSS levelling. Global models are truncated with steps of 10 degrees starting from degree and order 10 to degree and order 300. Omission error is computed from the degree of truncation to degree and order 2190 from EGM2008 and from the ERTM2160 topographic gravity field model for degrees above 2160.

different quality levels, which strongly depends on the instruments, the measurement procedure and the observer carefulness. In general, the results for most of the comparisons show similar behavior, but at different levels of accuracy. We have selected representative data sets from different continents and of different quality levels and validated the model derived geoid heights against these data sets. Here we show the results for Brazil, Germany, Japan and the United States (see acknowledgement).

     Figure 10 shows the results for the validation of the global models against the independently observed geoid heights at

GNSS levelling points for the four selected countries. Before drawing any conclusion from the results, the figure shall be explained shortly. Each tested model was truncated at steps of degree 10 of the spherical harmonic series starting from degree





and order 10 until 300. For each area the model geoid heights were computed up to the degree and order given on the $x$-axis and differences to the observed geoid heights were calculated. The omission error was computed separately from the EGM2008 and ERTM2160 models (see above) starting from the truncation degree to highest possible resolution and subtracted from the geoid differences. Finally, a planar correction surface was calculated for each area for the geoid differences and subtracted. From the remaining differences the RMS values shown on the $y$-axis were computed. In general, the lowest level of the RMS values is a good indicator about the quality of the GNSS levelling derived geoid heights as one can assume that the global satellite-only models perform similar over the whole globe.

Regarding this it becomes obvious that the German data set seems to be the best ground data set, while the Brazilian data set in average suffers from some inaccuracies, which are due to the size of the country and other geodetic infrastructure weaknesses. For Japan and the United States the RMS of differences is at a level of $7.5\,\mathrm{cm}$ and $9.5\,\mathrm{cm}$ respectively. The general shape of the curves is completely different for Brazil and the other data sets under investigation. The reason behind is that the more information from the EGM2008 model (without GOCE data) is used for computing the omission error the worse the result of the differences is. This indicates that GOCE data can improve significantly the overall performance of the global gravity field models in case no high quality ground data are available. In the other areas we consider here high quality ground data were used in EGM2008 and therefore it performs very good when most of the signal is computed from this model (i.e. for low truncation degrees). Looking to the different satellite-only solutions one can identify the limit of a pure GRACE solution (ITSG-Grace2018s) as the RMS of the differences starts to increase at lower degrees than for the other models. Including GOCE (and more GRACE) data significantly improve the higher resolution terms of the satellite-only models as it can be seen from the series from GOCO01s to GOCO06s where the higher release number means more satellite information included. When comparing the satellite-only models to the combined XGM2019e model the degree where they start to diverge somehow represent the maximum signal content of the satellite-only models, or in other words up to what degree and order such a model contains the full gravity field signal. Depending on the area we can identify that around degree 200 the models start to diverge. For Germany, where we probably have the best ground data set, the satellite contribution is superior may be up to degree 180. Regarding the GOCO06s model we can state from these comparisons that it is a state-of-the-art satellite-only model and that it performs best together with the GOCE-DIR6 and GOCE-TIM6 models, which on the higher end of the spectrum are all based on the same complete GOCE data set.

## 4.2  Orbit residuals

To evaluate the long-wavelength part of GOCO06s, we analyze orbit residuals between integrated dynamic orbits and GPS derived kinematic orbits of 4 LEO satellites of various altitudes and inclinations. The used missions with corresponding altitude and inclination are GOCE ($\approx 250\,\mathrm{km}$, 96.7°), TerraSAR-X (Buckreuss et al., 2003, $\approx 515\,\mathrm{km}$, 98°), Jason-2 (Neeck and Vaze, 2008, $\approx 1330\,\mathrm{km}$, 66°), and GRACE-B ($\approx 450\,\mathrm{km}$, 89°). We realize that this is not a fully independent evaluation since GOCO06s contains kinematic orbit data from the LEO satellites considered here. However, since the overall contribution of the GPS positions to the gravity field is very low (see Section 3.2), we argue that this approach still gives valid conclusions.





Furthermore, we used the kinematic GOCE and GRACE orbits of the Astronomical Institute at Bern University (Bock et al., 2014; Meyer et al., 2016) and not the in-house positions used in the gravity field recovery process.

First, a dynamic orbit for each satellite is integrated based on a fixed set of geophysical models, where only the static gravity field is substituted. The models used can be found in Table 2. We included trend and annual gravity field variations from

**Table 2.** Background models used in the dynamic orbit integration.

| Force | Model | Maximum degree |
|---|---|---|
| Earth's gravity field | estimated from GRACE solutions | |
| (annual/secular variations) | (CSR RL06, DDK4 filter applied) | 96 |
| Non-tidal ocean/atmosphere variation | AOD1B RL06 | 180 |
| Ocean Tides | FES2014b | 180 |
| Astronomical tides | IERS2010, JPL DE421 | N/A |
| Solid earth tides | IERS2010 | 4 |
| Atmospheric tides | AOD1B RL06 | 180 |
| Pole tides | IERS2010 (linear mean pole) | $c_{21}, s_{21}$ |
| Ocean pole tides | Desai (2002, linear mean pole) | 220 |
| Relativistic corrections | IERS2010 | - |
| Nonconservative forces | measured if available, else | |
| | JB2008 (drag), Knocke et al. (1988)/CERES (albedo), | |
| | Lemoine et al. (2013) (solar radiation pressure) | - |

GRACE to make sure all compared gravity fields relate to the same reference epoch.

Since satellites are not only affected by gravitational forces, but also non-conservative forces such as atmospheric drag, solar radiation pressure, and earth radiation pressure. Both GOCE and GRACE are equipped with accelerometers which measure the impact of these forces. The other satellites (Jason-2, TerraSar-X), do not feature such an instrument, therefore we make use of models to compute the effect. Specifically, we model the impact of atmospheric drag (Bowman et al., 2008), Earth's albedo (Knocke et al., 1988) and solar radiation pressure (Lemoine et al., 2013). Even though the quality of accelerometer

measurements is generally superior to the model output, sensor specific errors such as bias and scale need to be considered. Consequently we model and estimate both bias and scale as well as empirical parameters for both accelerometers and model output to reduce the potential impact of mismodelling and sensor errors on our evaluation. The values of these calibration parameters are estimated together with the initial state when the dynamic orbit is fitted to the kinematic satellite positions. One constant bias for each axis and a single full scale matrix (Klinger and Mayer-Gürr, 2016) is estimated per each 1-day arc for

all satellites. For Jason-2 and TerraSAR-X we additionally estimate a once-per-revolution bias to further reduce the impact of non-conservative forces. The metric we are using for the evaluation is root-mean-square (RMS) of the orbit differences between the integrated dynamic orbit and the purely GPS derived kinematic orbit positions. Epochs where the difference exceeds 15 cm





**Table 3.** RMS of differences between integrated dynamic and GPS derived kinematic orbits in centimeters for four selected months during the GOCE measurement time span for satellites of different altitude and inclination.

|           | GOCO05s | GOCO06s | GOCE TIM6 | GOCE DIR6 |
|-----------|---------|---------|-----------|-----------|
| GOCE      | 2.39    | 2.36    | 3.54      | 2.39      |
| TerraSAR-X| 5.66    | 5.65    | 5.73      | 5.67      |
| Jason-2   | 3.89    | 3.88    | 4.00      | 3.88      |
| GRACE-B   | 2.41    | 2.24    | 5.08      | 2.25      |

**Table 4.** RMS of differences between integrated dynamic orbit and GPS derived kinematic positions for GRACE-C January 2020 in centimeters.

|         | EIGEN-GRGS RL04 | GOCO06s | GOCO05s | ITSG-Grace op |
|---------|-----------------|---------|---------|---------------|
| GRACE-C | 3.06            | 2.44    | 2.60    | 2.32          |

are treated as outliers and subsequently removed. We randomly selected four months over the GOCE measurement time span (2009-11, 2010-03, 2011-10, and 2012-03).

We compare GOCO06s with its predecessor GOCO05s and other recent static gravity field models (see Fig. 8) models available on ICGEM (Ince et al., 2019). The results are summarized in Table 3. From the computed RMS values we can conclude that the most recent combination models (GOCO06s and DIR6) perform nearly equally well, with GOCO06s having a slight edge. Furthermore, we can see that there is a quality jump from the previous release GOCO05s. The GOCE-only model TIM6 performs worst for all satellites which highlights the importance of GRACE for stabilizing the long to medium

wavelengths and the polar gap. The overall larger RMS values for TerraSAR-X and Jason-2 reflect the challenges in modelling non-conservative forces for these satellites. Combined with the higher altitude, the contrast between the individual static gravity field solutions becomes very small. Still, we observe the same tendencies as for GOCE and GRACE.

Next to the validation of static gravity fields, orbit residuals are also very useful in evaluating the temporal constituents of gravity field solutions. We gauge the quality of the co-estimated temporal constituents of GOCO06s by integrating a dynamic

orbit arcs for GRACE Follow-On 1 (GRACE-C). We compare GOCO06s to EIGEN-GRGS RL04, a recent gravity field model which also features a time-variable part, the previous solution GOCO05s, and a GRACE-FO monthly solution. Instead of the trend and annual variation estimated from CSR RL06, we use the temporal constituents of the gravity fields to be compared. Both GOCO06s and GOCO05s provide secular and annual variations while EIGEN-GRGS RL04 has secular, annual and semi-annual variations estimated for shorter intervals. We perform the evaluation for January 2020, which is well beyond the

GRACE measurement time span. This choice is deliberate to assess the extrapolation capabilities of the compared gravity fields. The computed RMS values can be found in Table 4. We can see that GOCO06s outperforms both its predecessor and EIGEN-GRGS RL04 while unsurprisingly, the GRACE-FO monthly solution performs best.



## 5 Conclusions

The satellite-only gravity field model GOCO06s provides a consistent combination of spaceborne gravity observations from
a variety of satellite missions and measurement techniques. Each component of the solution was processed using state-of-the-
art methodology which results in a clear increase in the solution quality compared to the preceding GOCO solutions and the
individual input models. All contributing data sources were combined on the basis of full normal equations, with the individual
weights being determined by VCE. The long to medium spatial wavelengths covered by GOCO06s are mainly determined by
GRACE due to the high sensitivity of the intersatellite ranging observation to this frequency band. SLR primarily contributes to
degree 2, while the kinematic LEO orbits mainly contribute to the sectorial coefficients up to degree and order 150. Finally, the
medium to short wavelengths of the solution, starting from degree 120, are dominated by the GOCE gradiometer observations.
To reduce the energy in the higher spherical harmonic degrees, a Kaula-type regularization was applied from degree 151 to
300. The complementary nature of the used data mitigates weaknesses of the individual observation types, thus providing the
highest accuracy throughout the spatial frequency band covered by the solution.

Since an appropriate stochastic observation model is used during processing, GOCO06s exhibits realistic formal errors,
which alleviates further combination with additional, for example, terrestrial gravity field observations (Pail et al., 2016, 2018).
This extremely useful information is provided to the community in the form of the full system of unconstrained normal equa-
tions. All data products are published in well established file formats such as the ICGEM format for potential coefficients
(Barthelmes and Förste, 2011) and the SINEX format (IERS, 2006) for systems of normal equations.

## 6 Data availability

The primary model data (Kvas et al., 2019b) consisting of potential coefficients representing Earth's static gravity field, to-
gether with secular and annual variations are available on ICGEM (Ince et al., 2019). This data set is identified with the DOI
10.5880/ICGEM.2019.002.

Supplementary material consisting of the full variance-covariance matrix of the static potential coefficients and estimated
co-seismic mass changes are available on ifg.tugraz.at/GOCO.

*Author contributions.* Andreas Kvas computed the GRACE and LEO normal equations, developed and implemented the regularization
strategies for the temporal constituents and earthquakes, performed the combination of all individual data contributions, performed the orbit
validation and partially wrote the manuscript. Jan Martin Brockmann computed the GOCE SGG system of normal equations and partially
wrote the manuscript. Till Schubert and Wolf-Dieter Schuh contributed to the GOCE processing. Sandro Krauss computed and provided the
systems of normal equations for all SLR satellites. Thomas Gruber performed the model validation with in-situ GNSS leveling data data.
Ulrich Meyer provided the kinematic orbits of GRACE and GOCE used in the validation. Torsten Mayer-Gürr, Adrian Jäggi, Wolf-Dieter
Schuh, and Roland Pail acted as scientific advisors. All authors commented on the draft of the manuscript and on the discussion of the results.



*Competing interests.* The author declare no competing interests.

*Acknowledgements.* Parts of this work was financially supported by ESA GOCE HPF (main contract No. 18308/04/NL/MM). JMB and WDS
gratefully acknowledge the Gauss Centre for Supercomputing e.V. (www.gauss-centre.eu) for funding this study by providing computing time
through the John von Neumann Institute for Computing (NIC) on the GCS Supercomputer JUWELS at Jülich Supercomputing Centre (JSC).

Precise orbit determination of SLR satellites was accomplished with the software package GEODYN, kindly provided by the NASA
Goddard Space Flight Center.

GNSS levelling data sets applied in this study have been kindly provided by the Brazilian Institute of Geography and Statistics - IBGE,
2019 (Brazil), by the GeoBasis-DE / Geobasis NRW, 2018 (Germany), by the Japanese Geographical Survey Institute, 2003 (Japan) and by
the National Geodetic Survey, 2012 (USA). The authors are grateful to the institutions who made available the GNSS-levelling data as they
provide a unique reference for validating global gravity field models.

The authors gratefully acknowledge the ISDC at the German Research Centre for Geosciences for providing Champ, TerraSAR-X and
TanDEM-X data, the European Space Agency for providing GOCE and SWARM data, and Aviso for providing Jason data and the Interna-
tional Laser Ranging Service for providing Satellite laser ranging measurements.






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
