# Peer review of "GOCO06s - A satellite-only global gravity field model"

_Earth System Science Data, 2020_

## Referee Comment (RC1) · Anonymous Referee #1 · 29 Sep 2020

Review about the paper

**GOCO06s – A satellite-only global gravity field model**

submitted to Earth System Science Data (https://doi.org/10.5194/essd-2020-192)

Authors: Andreas Kvas, Jan Martin Brockmann, Sandro Krauss, Till Schubert, Thomas Gruber, Ulrich Meyer, Torsten Mayer-Gürr, Wolf-Dieter Schuh, Adrian Jäggi, and Roland Pail

**General remarks:**

This paper describes the generation and the characteristics of the satellite-only gravity field model GOCO06s. The computation of this model was done in a joint work by the partners of the GOCO consortium. GOCO06s has been already published last year for the user community on the ICGEM data base and on the GOCO webpages. This paper now accompanies the publication of the GOCO06s data set and will help users to understand the background of this model. GOCO06s is one of the best global satellite-only gravity field models currently available and it benefits especially from the matter of fact that data from practically all recent gravity satellite missions were included and combined via a refined combination technique. I appreciate especially that the authors published along with the model very valuable supplementary material comprising the full variance-covariance matrix of the static spherical harmonic coefficients and estimated co-seismic mass changes. The content and the outline of this paper is convincing for me. The paper is written in good English wording. I recommend only minor revision according to my specific remarks below:

**Specific remarks:**

**Page 2, lines 34/35:**
- The statement "Starting with the dedicated CHAllenging Minisatellite Payload satellite mission … different LEO satellites were tracked by GPS" is not 100% correct. There were LEO satellite missions before CHAMP with GPS receivers onboard, for instance Topex/Poseidon. CHAMP was the first satellite gravity mission with a GPS receiver onboard. Please modify this statement accordingly.
- Please correct this typo: **Reigbar** et al. 1999 → **Reigber** et al. 1999, please correct this name also in the reference list.

**Page 2, line 48:**
The reference Battrick 1999 is right. Nevertheless, I suggest to give a more recent reference in addition to Battrick 1999, for instance Floberghagen et al. 2011:
Rune Floberghagen, Michael Fehringer, Daniel Lamarre, Danilo Muzi, Björn Frommknecht, Christoph Steiger, Juan Piñeiro, Andrea da Costa: Mission design, operation and exploitation of the gravity field and steady-state ocean circulation explorer mission, J Geod (2011) 85:749–758, DOI 10.1007/s00190-011-0498-3

**Page 3, line 83 – 85:**

The statement "Basically, GRACE normal equations processed by GFZ (Dahle et al., 2019) are combined with GOCE normal equations assembled with the so-called direct approach (Bruinsma et al., 85 2014; Pail et al., 2011) and SLR normal equations." is not fully correct, since some of the GRACE and SLR normal equations used for the DIR-models were generated by CNES/GRGS. I suggest to modify this sentence as follows:

**Basically, GRACE normal equations processed by CNES/GRGS (Bruinsma et al. 2010) or GFZ (Dahle et al., 2019) are combined with GOCE normal equations assembled with the so-called direct approach (Bruinsma et al., 2014; Pail et al., 2011) and SLR normal equations."**

The additional reference is:
Bruinsma, S. L., J. M. Lemoine, R. Biancale, and N. Vales (2010), CNES/GRGS 10-day gravity field models (release 2) and their evaluation, Adv. Space Res., 45(4), 587–601, doi:10.1016/j.asr.2009.10.012

**Sections 2.2 and 2.3:**

I see a contradiction regarding GRACE between the sections 2.2. and 2.3:

In section 2.2 the GRACE contribution is described. Here among others it is said: "The normal equations feature a static part parametrized up to degree/order **200** and **secular and annual variations up to degree/order 120**…"

In contrast, in section 2.3 GRACE is mentioned again, but here as one of the LEO satellites and you write: "For CHAMP, GRACE, TerraSAR-X, TanDEM-X and SWARM we set up the normal equations up to degree and order **120**…" and "For all LEO satellites, **only the static** gravity field was modeled…"
I think, GRACE is mentioned in section 2.3 by mistake. Therefore, you should remove GRACE from section 2.3. Then, in section 2.2 some sentences about the used GRACE kinematic orbit positions should be added, similar to such for the other LEOs in lines 143 – 145.

**Section 2.2, 2.3 and 2.4:**
You estimated temporal gravity variations for GRACE and SLR only. But its known that other satellites like CHAMP, GOCE (here the long wavelengths only based on the kinematic orbits) and Swarm were also sensitive for temporal gravity field variations. You mentioned this in principle in the introduction. Do you think it makes sense to include both satellites here to improve the estimates of the long wavelengths, for instance in a future upgrade of GOCEO06s? Could you please add some sentences about this question e.g. in section 2.3 or in the introduction?

**Page 7, line 174:**

I suggest to give the obtained relative weights. This would be of interest to others users which are processing SLR data.

**Page 8, formulae (11) and (12):**
Could you please also give the relative weights applied for GOCE, GRACE and the LEO satellites?

**Page 8, lines 201/202**

You said, "… the SLR system of normal equations was artificially down weighted by a factor of 10 in each iteration step." Did you try other factors for down weighting? If you used only one factor you can never be sure that your empirical down weighting factor is optimum. Please comment.

**Page 15, line 320 and Figure 7:**

"… evaluated close to the epicenter of the 2004 Indian Ocean earthquake…" Please give the location of this point. Is it east or west of the fault line? Please give the precise geographical coordinates ore indicate the position in the map of figure 6a.

**Figure 7:**

The caption for this figure is inaccurate since the location is not mentioned. I suggest to write: "Comparison of estimated secular variation from GOCO05s, GOCO06s (including estimated co-seismic mass change) and filtered GRACE monthly solutions in terms of equivalent water height (EWH) **for a location close to the epicenter of the 2004 Indian Ocean earthquake (c.f. figure 6a)**

**Page 16, lines 331 – 341**

Your discussion about figure 8 is partially not clear for me:

You statement "The other large differences between the compared models in degrees below 60 are primarily explained by the respective reference epochs (for example, 2010-01-01 for GOCO06s and 2010-09-01 for GOCE DIR6)" is a little bit misleading: Even the spectra of Goco06s and GOCE DIR6 are closely together below degree 60. Please discuss e.g. the difference between GOCEO06s and GOCE TIM5 which is significantly larger.

Your statement "Concerning the GOCE gradiometer reprocessing we can see improvements from degree 150, where these data start to dominate the solutions…." is unclear for me, since the difference spectra of all five compared models are almost the same. Please explain better what you mean.

**Page 17, figures 8 and 9:**

- In the captions of both figures it's said that "8 degree polar cap (are) excluded" in the degree variances. This statement is unclear for me. Does it mean, that you start to exclude polar gap components beyond degree 8 up to the maximum degree? What is in detail excluded, some low order coefficients? Please explain. I suggest to explain the exclusion of the polar gaps for the spectra within the text of chapter 4 and not in the captions.

- Beside of the polar gap comment I ask you to modify the captions as follows:

  **Figure 8: Difference degree amplitudes of various state-of-the-art satellite-only models to the combined model XGM2016**

  **Figure 9: Difference degree amplitudes to GOCO06s (solid lines) and the corresponding formal errors (dashed lines) for the individual GOCO06s components…**

**Page 18, lines 372 – 374:**

"In addition, an independent GRACE/GOCE satellite-only combination model namely GOCE-DIR6 is used for comparisons, which to a large extent is based on the same amount of GRACE and GOCE data as GOCO06s (Förste et al., 2019)"

This sentence is inaccurate and the place of the mentioned reference is disadvantageo. I suggest to rewrite as followes:

**"In addition, an independent SLR/GRACE/GOCE satellite-only combination model namely GOCE-DIR6 (Förste et al., 2019) is used for comparisons, which to a large extent is based on the same amount of SLR, GRACE and GOCE data as GOCO06s."**

**Page 2, Table 2:**

Which maximum degree was used for the various Earth's gravity field models? Please give the number. It should be the same in all cases. If not, please discuss.

The item "annual/secular variations" and the related model is given in brackets there. Why?

---

## Referee Comment (RC2) · Anonymous Referee #2 · 1 Dec 2020

The paper presents the latest global gravity field model computed by the GOCO ("Gravity Observation Combination") project, namely the GOCO06s model based on over a billion observations acquired over 15 years from 19 satellites with different complementary observation principles. This means that all recent gravity mission data have been combined and included in the model, making it one the best satellite-only gravity field models available. The primary data of GOCO06 are the geo-potential coefficients of the spherical harmonic series representing the Earth static gravity field, together with secular and annual variations. These data are available through the ICGEM web site. Other data accompanying the model are the coefficients of the full variance-covariance matrix of the static potential coefficients and the estimated co-seismic mass changes; these data are available through the IFG Graz web site. The paper provides valuable

information for the users of the GOCO06s model, in terms of understanding both the information content and the computational strategy applied to combine the available data in the model. Besides, the paper is very well written, in good English form. Having also read the general and specific remarks and questions of Referee #1 I cannot but agree. Moreover, in my opinion the Authors have replied in a satisfactory way to all comments, explaining and rephrasing sentences where needed and adding appropriate references as requested. Therefore I have nothing else to add or request, and I recommend publication of the paper in its revised form after the comments of Referee #1.
* * *

---

## Author Response (AR1)

The paper presents the latest global gravity field model computed by the GOCO ("Gravity Observation Combination") project, namely the GOCO06s model based on over a billion observations acquired over 15 years from 19 satellites with different complementary observation principles. This means that all recent gravity mission data have been combined and included in the model, making it one the best satellite-only gravity field models available. The primary data of GOCO06 are the geo-potential coefficients of the spherical harmonic series representing the Earth static gravity field, together with secular and annual variations. These data are available through the ICGEM web site. Other data accompanying the model are the coefficients of the full variance-covariance matrix of the static potential coefficients and the estimated co-seismic mass changes; these data are available through the IFG Graz web site. The paper provides information for the users of the GOCO06s model, in terms of understanding both the information content and the computational strategy applied to combine the available data in the model. Besides, the paper is very well written, in good English form. Having also read the general and specific remarks and questions of Referee #1 I cannot but agree. Moreover, in my opinion the Authors have replied in a satisfactory way to all comments, explaining and rephrasing sentences where needed and adding appropriate references as requested. Therefore I have nothing else to add or request, and I recommend publication of the paper in its revised form after the comments of Referee #1.

**AC:** We thank the reviewer for their careful reading of the manuscript and the constructive remarks. Since the points addressed are in line with Anonymous Referee #1, we refer here to the point-by-point response given to Anonymous Referee #1 on the following pages.

**Anonymous Referee #1**

Review about the paper

**GOCO06s – A satellite-only global gravity field model**

submitted to Earth System Science Data (https://doi.org/10.5194/essd-2020-192)

Authors: Andreas Kvas, Jan Martin Brockmann, Sandro Krauss, Till Schubert, Thomas Gruber, Ulrich Meyer, Torsten Mayer-Gürr, Wolf-Dieter Schuh, Adrian Jäggi, and Roland Pail

**General remarks:**

This paper describes the generation and the characteristics of the satellite-only gravity field model GOCO06s. The computation of this model was done in a joint work by the partners of the GOCO consortium. GOCO06s has been already published last year for the user community on the ICGEM data base and on the GOCO webpages. This paper now accompanies the publication of the GOCO06s data set and will help users to understand the background of this model. GOCO06s is one of the best global satellite-only gravity field models currently available and it benefits especially from the matter of fact that data from practically all recent gravity satellite missions were included and combined via a refined combination technique. I appreciate especially that the authors published along with the model very valuable supplementary material comprising the full variance-covariance matrix of the static spherical harmonic coefficients and estimated co-seismic mass changes. The content and the outline of this paper is convincing for me. The paper is written in good English wording. I recommend only minor revision according to my specific remarks below:

**AC:** We thank the reviewer for their insightful comments, which helped us to identify parts in need of clarification and undoubtedly allowed us to improve the quality of the manuscript. Below is the point-by-point response to the specific remarks.

**Page 2, lines 34/35:**
- The statement "Starting with the dedicated CHAllenging Minisatellite Payload satellite mission … different LEO satellites were tracked by GPS" is not 100% correct. There were LEO satellite missions before CHAMP with GPS receivers onboard, for instance Topex/Poseidon. CHAMP was the first satellite gravity mission with a GPS receiver onboard. Please modify this statement accordingly.
- Please correct this typo: **Reigbar** et al. 1999 → **Reigber** et al. 1999, please correct this name also in the reference list.

**AC:** We rephrased the sentence to:

"The CHAllenging Minisatellite Payload satellite mission (CHAMP, Reigber 1999) was the first dedicated gravity field mission tracked by GPS. Since then multiple other dedicated and non-dedicated satellites equipped with GPS receivers have been used derive static and temporal gravity field models of the Earth (…) in this so called satellite-to-satellite tracking in high-low (SST-hl) configuration."

To correctly state that there were GPS-tracked satellites before CHAMP and fixed the typographical error in the reference.
* * *
**Page 2, line 48:**

The reference Battrick 1999 is right. Nevertheless, I suggest to give a more recent reference in addition to Battrick 1999, for instance Floberghagen et al. 2011:
Rune Floberghagen, Michael Fehringer, Daniel Lamarre, Danilo Muzi, Björn Frommknecht, Christoph Steiger, Juan Piñeiro, Andrea da Costa: Mission design, operation and exploitation of the gravity field and steady-state ocean circulation explorer mission, J Geod (2011) 85:749– 758, DOI 10.1007/s00190-011-0498-3

**AC:** We added the more recent reference to Floberhagen et al. (2011) in addition to Battrick (1999).
* * *
**Page 3, line 83 – 85:**

The statement "Basically, GRACE normal equations processed by GFZ (Dahle et al., 2019) are combined with GOCE normal equations assembled with the so-called direct approach (Bruinsma et al., 85 2014; Pail et al., 2011) and SLR normal equations." is not fully correct, since some of the GRACE and SLR normal equations used for the DIR-models were generated by CNES/GRGS. I suggest to modify this sentence as follows:

**Basically, GRACE normal equations processed by CNES/GRGS (Bruinsma et al. 2010) or GFZ (Dahle et al., 2019) are combined with GOCE normal equations assembled with the so-called direct approach (Bruinsma et al., 2014; Pail et al., 2011) and SLR normal equations."**

The additional reference is:
Bruinsma, S. L., J. M. Lemoine, R. Biancale, and N. Vales (2010), CNES/GRGS 10-day gravity field models (release 2) and their evaluation, Adv. Space Res., 45(4), 587–601, doi:10.1016/j.asr.2009.10.012

**AC:** We rephrased the sentence and added the appropriate reference.
* * *
**Sections 2.2 and 2.3:**

I see a contradiction regarding GRACE between the sections 2.2. and 2.3:

In section 2.2 the GRACE contribution is described. Here among others it is said: "The normal equations feature a static part parametrized up to degree/order **200** and **secular and annual variations up to degree/order 120**…"

In contrast, in section 2.3 GRACE is mentioned again, but here as one of the LEO satellites and you write: "For CHAMP, GRACE, TerraSAR-X, TanDEM-X and SWARM we set up the normal equations up to degree and order **120**…" and "For all LEO satellites, **only the static** gravity field was modeled…"

I think, GRACE is mentioned in section 2.3 by mistake. Therefore, you should remove GRACE from section 2.3. Then, in section 2.2 some sentences about the used GRACE kinematic orbit positions should be added, similar to such for the other LEOs in lines 143 – 145.

**AC:** Thank you for this observation, indeed the parametrization of the GRACE LEO orbits was not properly presented. To provide a clear picture, we amended the introduction to section 2.3.
* * *
**Section 2.2, 2.3 and 2.4:**

You estimated temporal gravity variations for GRACE and SLR only. But its known that other satellites like CHAMP, GOCE (here the long wavelengths only based on the kinematic orbits) and Swarm were also sensitive for temporal gravity field variations. You mentioned this in principle in the introduction. Do you think it makes sense to include both satellites here to improve the estimates of the long wavelengths, for instance in a future upgrade of GOCEO06s? Could you please add some sentences about this question e.g. in section 2.3 or in the introduction?

**AC:** The decision to not model temporal variations for the LEO orbits was made from the perspective of computational demands vs. impact on the solution. As you rightly state, the kinematic orbits provide information for temporal gravity field recovery and with an ever-growing data record through currently active and planned missions it is an interesting observation set. It is therefore certainly worth revisiting the inclusion of temporal variations from LEO satellites in future reprocessings.

We added the following sentences to clarify our motivation to not include temporal variations for the kinematic LEO orbits:

"We made the decision to not include secular and annual variations in the parameter set based on practical considerations. Even though SST-hl has skill in determining the time-variable gravity field, the expected contribution of the LEO orbits to the temporal constituents, with the presence of GRACE intersatellite-ranging observations and SLR, will be low. This results in a steep trade-off between the increase in solution quality and the computational demands required to set up the additional parameters. "
* * *
**Page 7, line 174:**

I suggest to give the obtained relative weights. This would be of interest to others users which are processing SLR data.

**AC:** We added a Table with the obtained weights for the 10 processed satellites.

**Page 8, formulae (11) and (12):**

Could you please also give the relative weights applied for GOCE, GRACE and the LEO satellites?

**AC:** The GOCE, GRACE and LEO orbit contributions to GOCO06s feature a sophisticated stochastic model of the input observables. This means that the resulting normal equations already properly describe the accuracy with which each technique can determine the potential coefficients in an absolute sense. The weights determined in the VCE therefore do deviate from 1 only in the magnitude of $10^{-4}$ even without any pre-scaling. We added the sentence:

"Consequently, the weights of these contributions remain close to 1 throughout the VCE iterations."

to better reflect this circumstance.
* * *
**Page 8, lines 201/202**

You said, "… the SLR system of normal equations was artificially down weighted by a factor of 10 in each iteration step." Did you try other factors for down weighting? If you used only one factor you can never be sure that your empirical down weighting factor is optimum. Please comment.

**AC:** We determined the factor by analyzing the differences between the SLR solution and GOCO05s (which also contains an SLR solution). The gravity field signal cancels out and the remaining differences contain the noise of both solutions. This indicates the order of magnitude of the true errors compared to the formal errors. We agree that such a manual weighting scheme is not ideal and might result in a sub-optimal solution in a statistical sense. Ideally, one would model the normal point observation noise as it is done in the GRACE, GOCE and LEO contributions, which is something worth revisiting in future iterations.
* * *
**Page 15, line 320 and Figure 7:**

"… evaluated close to the epicenter of the 2004 Indian Ocean earthquake…" Please give the location of this point. Is it east or west of the fault line? Please give the precise geographical coordinates ore indicate the position in the map of figure 6a.

**AC:** We added the geographic coordinates to the Figure caption and additionally included an overview map with the point indicated to the Figure itself.
* * *
**Figure 7:**

The caption for this figure is inaccurate since the location is not mentioned. I suggest to write: "Comparison of estimated secular variation from GOCO05s, GOCO06s (including estimated co-seismic mass change) and filtered GRACE monthly solutions in terms of equivalent water height (EWH) **for a location close to the epicenter of the 2004 Indian Ocean earthquake (c.f. figure 6a)**

**AC:** Please see the response to the previous remark.

**Page 16, lines 331 – 341**

Your discussion about figure 8 is partially not clear for me:

You statement "The other large differences between the compared models in degrees below 60 are primarily explained by the respective reference epochs (for example, 2010-01-01 for GOCO06s and 2010-09-01 for GOCE DIR6)" is a little bit misleading: Even the spectra of Goco06s and GOCE DIR6 are closely together below degree 60. Please discuss e.g. the difference between GOCEO06s and GOCE TIM5 which is significantly larger.

**AC:** You are fully correct, not only the reference epoch but also the used observation techniques play a role in this frequency band. The GOCE TIM models rely on GOCE kinematic orbits and gradiometer observations only and therefore cannot reach the noise level of combination models which use GRACE intersatellite ranging observations, which are far more accurate in this frequency band. To correct for this oversight, we restructured this part of the manuscript to:

"The differences between the compared models in degrees below 60 are on the one hand explained by the respective reference epochs (for example, 2010-01-01 for GOCO06s and 2010-09-01 for GOCE DIR6) and, on the other hand, by the used observation techniques. GOCE TIM6 only relies on kinematic orbit data and gradiometer observations, which do not reach the superior accuracy of the GRACE intersatellite ranging observations in this frequency band."

Your statement "Concerning the GOCE gradiometer reprocessing we can see improvements from degree 150, where these data start to dominate the solutions…." is unclear for me, since the difference spectra of all five compared models are almost the same. Please explain better what you mean.

**AC:** To better highlight the differences between the models in the higher degrees, we added a second panel to Figure 8 which only shows this frequency band. Furthermore, we rephrased the corresponding text in the manuscript:

"In Figure 8(b), we can also identify differences between the solutions starting from degree 150. Here, we can clearly distinguish between models (GOCO06s, TIM6, DIR6) based on the latest reprocessing of GOCE gradiometer data (version tag 0202) and models based on the previous release (GOCO05s, EIGEN-GRGS RL04)."
* * *
**Page 17, figures 8 and 9:**

- In the captions of both figures it's said that "8 degree polar cap (are) excluded" in the degree variances. This statement is unclear for me. Does it mean, that you start to exclude polar gap components beyond degree 8 up to the maximum degree? What is in detail excluded, some low order coefficients? Please explain. I suggest to explain the exclusion of the polar gaps for the spectra within the text of chapter 4 and not in the captions.

- Beside of the polar gap comment I ask you to modify the captions as follows:

**Figure 8: Difference degree amplitudes of various state-of-the-art satellite-only models to the combined model XGM2016**

**Figure 9: Difference degree amplitudes to GOCO06s (solid lines) and the corresponding formal errors (dashed lines) for the individual GOCO06s components…**

**AC:** We clarified our intentions behind the use of the polar cap and added an appropriate reference. Specifically, we changed the caption text of Figure 8 to

"Difference degree amplitudes of various state-of-the-art satellite-only models to the combined model XGM2016 (polar cap with 8° aperture angle excluded)."

and added the following text to the manuscript:

"Because of the GOCE orbit inclination of 96.7°, no data is collected directly above the poles. This polar gap has a distinct mapping to certain low-order coefficients of the gravity field (Sneeuw and van Gelderen, 1997). Consequently, these coefficients are highly correlated and less accurately determined in GOCE-only models, such as GOCE TIM6, where no other gravity field information is used. To avoid these low-order coefficients dominating the degree amplitudes and to ensure a consistent comparison, we excluded the coefficients corresponding to a polar cap with 8° aperture angle in all models, according to the rule of thumb given in Sneeuw and van Gelderen (1997)."

In Figure 9, we changed the caption text according to your suggestion and added the same information about the polar cap as in Figure 8:

"Difference degree amplitudes to GOCO06s (solid lines) and corresponding formal errors (dashed lines) of the individual GOCO06s components (polar cap with 8° aperture angle excluded). SLR is not shown because no standalone solution can be computed due to the ill-posed adjustment problem."
* * *
**Page 18, lines 372 – 374:**

"In addition, an independent GRACE/GOCE satellite-only combination model namely GOCEDIR6 is used for comparisons, which to a large extent is based on the same amount of GRACE and GOCE data as GOCO06s (Förste et al., 2019)"

This sentence is inaccurate and the place of the mentioned reference is disadvantageo. I suggest to rewrite as followes:

**"In addition, an independent SLR/GRACE/GOCE satellite-only combination model namely GOCE-DIR6 (Förste et al., 2019) is used for comparisons, which to a large extent is based on the same amount of SLR, GRACE and GOCE data as GOCO06s."**

**AC:** We rephrased the sentence according to your suggestion.

**Page 2, Table 2:**

Which maximum degree was used for the various Earth's gravity field models? Please give the number. It should be the same in all cases. If not, please discuss.

**AC**: All evaluated gravity fields were used up to degree 200. We added this information into the force model table (now Table 3).

The item "annual/secular variations" and the related model is given in brackets there. Why?

**AC**: We added an indentation, so it becomes clear that this is a continuation of the previous table line.

**GOCO06s – A satellite-only global gravity field model**

[revised manuscript text omitted]
 except GRACE, only the static gravity field was modeled. We made the decision to not include secular and annual variations in the parameter set based on practical considerations. Even though SST-hl has skill in determining the time-variable gravity field, the expected contribution of the LEO orbits to the temporal constituents, with the presence of GRACE intersatellite-ranging observations and SLR, will be low. That results in a steep trade-off between the increase in solution quality and the computational demands required to set up the additional parameters. Since we applied an appropriate stochastic model in the assembly of the systems of normal equations, the relative weighting between the satellites and epochs is already contained in the normal equations. Therefore, we only need to accumulate all satellites and months, as

$$\mathbf{N}_{\text{LEO}} = \sum_k \sum_s \mathbf{N}_k^{(s)}, \qquad \mathbf{n}_{\text{LEO}} = \sum_k \sum_s \mathbf{n}_k^{(s)}. \tag{3}$$

The kinematic orbits provide gravity field information primarily to the lower (near-) sectorial coefficients of the spherical harmonic spectrum, to which the GRACE inter-satellite range-rates are less sensitive.

**2.4 Satellite Laser Ranging (SLR)**

To stabilize the long wavelength part of the spectrum, we added SLR observations to the combination. The used observations match the GRACE time span from 2002-04 to 2016-08 and feature a total of 10 satellites, including LAGEOS 1/2, Ajisai, Stella, Starlette, LARES, LARETS, Etalon 1/2 and BLITS. The SLR observations were processed in weekly batches consisting of three 7-day arcs and one arc of variable length to complete the month, resulting in systems of normal equations $\mathbf{N}_k^{(s)} \hat{\mathbf{x}}_k^{(s)} = \mathbf{n}_k^{(s)}$ up to d/o 60 for each satellite $s$. To obtain the normal equations for static, trend, and annual oscillation from the weekly systems of normal equations we first perform a parameter transformation. We express the weekly potential coefficients $\mathbf{x}_k^{(s)}$ in terms of static, trend and annual oscillation as

$$\mathbf{x}_k^{(s)} = \mathbf{x}_{\text{static}}^{(s)} + \mathbf{x}_{\text{trend}}^{(s)}(t_k - t_0) + \mathbf{x}_{\text{cos}}^{(s)} \cos(\omega(t_k - t_0)) + \mathbf{x}_{\text{sin}}^{(s)} \sin(\omega(t_k - t_0)), \tag{4}$$

or equivalently in matrix notation

$$\mathbf{x}_k^{(s)} = \underbrace{\begin{bmatrix} \mathbf{I} & \mathbf{I} \cdot (t_k - t_0) & \mathbf{I} \cdot \cos(\omega(t_k - t_0)) & \mathbf{I} \cdot \sin(\omega(t_k - t_0)) \end{bmatrix}}_{:=\mathbf{F}} \begin{bmatrix} \mathbf{x}_{\text{static}}^{(s)} \\ \mathbf{x}_{\text{trend}}^{(s)} \\ \mathbf{x}_{\text{cos}}^{(s)} \\ \mathbf{x}_{\text{sin}}^{(s)} \end{bmatrix}. \tag{5}$$

**Table 2.** Relative weights of SLR satellites determined through VCE.

| Satellite | $w^{(s)}$ | Satellite | $w^{(s)}$ |
|-----------|-----------|-----------|-----------|
| LAGEOS 1  | 41.9      | Etalon 1  | 1.0       |
| LAGEOS 2  | 38.3      | Etalon 2  | 1.6       |
| Ajisai    | 3.2       | LARES     | 1.1       |
| Stella    | 3.7       | Larets    | 3.2       |
| Starlette | 2.8       | BLITS     | 3.0       |

Here, $\mathbf{I}$ is an identity matrix of appropriate dimension, $t_0$ is the reference epoch of the combined gravity field and $\omega$ is the angular frequency corresponding to an oscillation of 365.25 days. Using this parameter substitution we can transform the weekly systems of normal equations and accumulate over whole time span,

$$\mathbf{N}_{\text{SLR}}^{(s)} = \sum_k \mathbf{F}^T \mathbf{N}_k^{(s)} \mathbf{F}, \qquad \mathbf{n}_{\text{SLR}}^{(s)} = \sum_k \mathbf{F}^T \mathbf{n}_k^{(s)}. \tag{6}$$

After the accumulation we determined the relative weight between the individual SLR satellites $s$ by variance component estimation (VCE). Since we processed all satellites consistently, it is reasonable to assume that the formal errors are underestimated in the same fashion for all SLR targets, even though no proper stochastic model for the SLR observation was used in the processing. To determine the relative weights $w^{(s)}$ of the satellites, we do not make use of the full system of normal equations, but only use the static part up to degree and order 10. In Table 2, the determined relative weights can be found. Note that these are specific to the processing applied here. Additionally, we applied a Kaula constraint to stabilize the system of equations. The final normal equation matrix and right-hand-side used in the combination procedure is then computed as the weighted sum of all satellites,

$$\mathbf{N}_{\text{SLR}} = \sum_s w^{(s)} \mathbf{N}^{(s)}, \qquad \mathbf{n}_{\text{SLR}} = \sum_s w^{(s)} \mathbf{n}^{(s)}. \tag{7}$$

**3 Combination process**

We combined the individual contributions from GOCE, GRACE, kinematic LEO orbits, SLR, and constraints on the basis of normal equations using VCE. VCE is a widely used technique in geodesy for combining different observation groups (Koch and Kusche, 2002; Lemoine et al., 2013; Meyer et al., 2019). The general idea is to determine the relative weights $w_*$ between heterogeneous observation types within the same least squares adjustment where the unknown parameters, in our case the gravity field, are estimated. Typically, this is an iterative procedure where initial weights are refined. The inverse of these weights, called variance factors, are the quotient of a residual square sum and the redundancy of the observation group. In the most general form this quotient is given as

$$\sigma_k^2 = \frac{\hat{\mathbf{e}}^T \mathbf{\Sigma}^{-1} \mathbf{V}_k \mathbf{\Sigma}^{-1} \hat{\mathbf{e}}}{\text{trace}\left\{ (\mathbf{\Sigma}^{-1} - \mathbf{\Sigma}^{-1} \mathbf{A}_k \bar{\mathbf{N}}^{-1} \mathbf{A}_k^T \mathbf{\Sigma}^{-1}) \mathbf{V}_k \right\}}, \tag{8}$$

where $\hat{\mathbf{e}}$ are post-fit residuals, $\boldsymbol{\Sigma} = \sum_k \sigma_k^2 \mathbf{V}_k$ is the compound covariance matrix and $\mathbf{A}_k$ is the design matrix of the $k$-th observation group. The iterative nature of VCE can be seen in Eq. (8) where the compound covariance matrix computed from the variance factors of the previous iteration is necessary to compute the new values. For observation groups which are given as normal equations, so all satellite contributions and also the Kaula constraint, this simplifies to

$$\sigma_k^2 = \frac{(\mathbf{l}^T \mathbf{P} \mathbf{l})_k - 2\mathbf{n}^T \hat{\mathbf{x}} + \hat{\mathbf{x}}^T \mathbf{N}_k \hat{\mathbf{x}}}{n - \text{trace}\left\{\bar{\mathbf{N}}^{-1} \mathbf{N}_k\right\} \cdot \sigma_k^{-2}}. \tag{9}$$

The general form presented in Eq. (8) comes into play when we determine the weights of the regionally varying constraints for the trend and annual signal (see Section 3.1). There we find that, for example, the variance factors for the trend constraint can be computed from

$$\sigma_k^2 = \frac{\hat{\mathbf{x}}_{\text{trend}}^T \boldsymbol{\Sigma}_\Omega^{-1} \mathbf{V}_k \boldsymbol{\Sigma}_\Omega^{-1} \hat{\mathbf{x}}_{\text{trend}}}{\text{trace}\left\{[\boldsymbol{\Sigma}_\Omega^{-1} - \boldsymbol{\Sigma}_\Omega^{-1}(\bar{\mathbf{N}}_{\text{trend}} + \boldsymbol{\Sigma}_\Omega^{-1})^{-1} \boldsymbol{\Sigma}_\Omega^{-1}] \mathbf{V}_k\right\}}. \tag{10}$$

Here, $\hat{\mathbf{x}}_{\text{trend}}$ are the estimated trend parameters and $\bar{\mathbf{N}}_{\text{trend}}$ is the accumulated system of normal equations of all satellites. The determination of the relative weights $w_* = \frac{1}{\sigma_*^2}$ is a key criteria for the overall solution quality. The GOCE, GRACE and kinematic orbit contributions feature a proper stochastic model of the used observables and therefore realistic formal errors. This is a prerequisite for a proper determination of relative weights using VCE (Meyer et al., 2019). Consequently, the weights of these contributions remain close to 1 throughout the VCE iterations. The SLR contribution used here lacks an adequate stochastic observation model which results in formal accuracies which are too optimistic. To counteract the high weights arising from this mismodeling, the SLR system of normal equations was artificially downweighted by a factor of 10 in each iteration step. The full, accumulated system of normal equations solved in each iteration step is given by

$$\bar{\mathbf{N}} = w_{\text{GOCE}} \mathbf{N}_{\text{GOCE}} + w_{\text{GRACE}} \mathbf{N}_{\text{GRACE}} + w_{\text{LEO}} \mathbf{N}_{\text{LEO}} + w_{\text{SLR}} \mathbf{N}_{\text{SLR}} + w_{\text{Kaula}} \mathbf{K} + \boldsymbol{\Sigma}_{\text{trend}}^{-1}(\mathbf{w}_{\text{trend}}) + \boldsymbol{\Sigma}_{\text{annual}}^{-1}(\mathbf{w}_{\text{annual}}), \tag{11}$$

$$\bar{\mathbf{n}} = w_{\text{GOCE}} \mathbf{n}_{\text{GOCE}} + w_{\text{GRACE}} \mathbf{n}_{\text{GRACE}} + w_{\text{LEO}} \mathbf{n}_{\text{LEO}} + w_{\text{SLR}} \mathbf{n}_{\text{SLR}}. \tag{12}$$

It features all satellite contributions as well as constraints for certain parameters. In order to reduce the power in the higher spherical harmonic degrees and specifically the polar regions where the GOCE gradiometer observation provide little to no information, the solution is zero-constrained using a Kaula-type signal model for degrees higher than 150. This Kaula regularization is represented by the matrix $\mathbf{K}$. Furthermore, the co-estimated trend and annual oscillation are also zero-constrained with a regionally varying regularization matrix $\boldsymbol{\Sigma}_{\text{trend}}^{-1}$ and $\boldsymbol{\Sigma}_{\text{annual}}^{-1}$ respectively. The vectors $\mathbf{w}_{\text{trend}}$ and $\mathbf{w}_{\text{
[revised manuscript text omitted]
.  Because of the GOCE orbit inclination of $96.7°$, no data is collected directly above the poles. This polar gap has a distinct mapping to certain low-order coefficients of the gravity field (Sneeuw and van Gelderen, 1997). Consequently, these coefficients are highly correlated and less accurately determined in GOCE-only models, such as GOCE TIM6, where no other gravity field information is used. To avoid these low-order coefficients dominating the degree amplitudes

[Figure]

**Figure 8.**  Difference degree amplitudes of various state-of-the-art satellite-only models  to the combined model XGM2016 ( polar cap with 8° aperture angle excluded). Panel (a) shows the whole spherical harmonic spectrum covered by the models, while panel (b) only shows degrees 150 to 300.

and to ensure a consistent comparison, we excluded the coefficients corresponding to a polar cap with 8° aperture angle in all models, according to the rule of thumb given in Sneeuw and van Gelderen (1997). The differences between the compared models in degrees below 60 are  on the one hand explained by the respective reference epochs (for example, 2010-01-01 for GOCO06s and 2010-09-01 for GOCE DIR6)

355  and, on the other hand, by the used observation techniques. GOCE TIM6 only relies on kinematic orbit data and gradiometer observations, which do not reach the superior accuracy of the GRACE intersatellite ranging observations in this frequency band.  In Figure 8(b), we can also identify differences between the solutions starting from degree 150. Here, we can clearly distinguish between models

360  (GOCO06s, TIM6, DIR6) based on the latest reprocessing of GOCE gradiometer data (version tag 0202) and models based on the previous release (GOCO05s, EIGEN-GRGS RL04). Figure 9 shows the degree amplitudes of component-wise differences with respect to the combined solution of each component of GOCO06s. We excluded SLR here because the very ill-posed system of normal equations can only be solved up to degree 5-6 without additional information (Cheng et al., 2011). It nicely summarizes the major contributions to the final GOCO06s solution and shows the consistency

365 of formal and empirical errors. It further highlights the importance of stochastic modeling which enables this consistency.

[Figure]

**Figure 9.**  Difference degree amplitudes to GOCO06s (solid lines) and corresponding formal errors (dashed lines) of the individual GOCO06s components ( polar cap with 8° aperture angle 
[revised manuscript text omitted]
$ 250 km, 96.7°), TerraSAR-X (Buckreuss et al., 2003, $\approx$515 km, 98°), Jason-2 (Neeck and Vaze, 2008, $\approx$1330 km, 66°), and GRACE-B ($\approx$450 km, 89°). We realize that this is not a fully independent evaluation since GOCO06s contains kinematic orbit data from the LEO satellites considered here. However, since the overall contribution of

440 the GPS positions to the gravity field is very low (see Section 3.2), we argue that this approach still gives valid conclusions. Furthermore, we used the kinematic GOCE and GRACE orbits of the Astronomical Institute at Bern University (Bock et al., 2014; Meyer et al., 2016) and not the in-house positions used in the gravity field recovery process.

First, a dynamic orbit for each satellite is integrated based on a fixed set of geophysical models, where only the static gravity field is substituted. The models used can be found in Table 3. We included trend and annual gravity field variations from

445 GRACE to make sure all compared gravity fields relate to the same reference epoch.

Since satellites are not only affected by gravitational forces, but also non-conservative forces such as atmospheric drag, solar radiation pressure, and earth radiation pressure. Both GOCE and GRACE are equipped with accelerometers which measure the impact of these forces. The other satellites (Jason-2, TerraSar-X), do not feature such an instrument, therefore we make

**Table 3.** Background models used in the dynamic orbit integration.

[revised manuscript text omitted]

Teixeira da Encarnação, J., Arnold, D., Bezděk, A., Dahle, C., Doornbos, E., van den IJssel, J., Jäggi, A., Mayer-Gürr, T., Sebera, J., Visser, P., and Zehentner, N.: Gravity Field Models Derived from Swarm GPS Data, Earth, Planets and Space, 68, 127, https://doi.org/10.1186/s40623-016-0499-9, 2016.

Teixeira da Encarnação, J., Visser, P., Arnold, D., Bezdek, A., Doornbos, E., Ellmer, M., Guo, J., van den IJssel, J., Iorfida, E., Jäggi, A., Klokocník, J., Krauss, S., Mao, X., Mayer-Gürr, T., Meyer, U., Sebera, J., Shum, C. K., Zhang, C., Zhang, Y., and Dahle, C.: Description of the multi-approach gravity field models from Swarm GPS data, Earth Syst. Sci. Data, 12, 1385–1417, https://doi.org/10.5194/essd-12-1385-2020, https://essd.copernicus.org/articles/12/1385/2020/https://essd.copernicus.org/articles/12/1385/2020/essd-12-1385-2020.pdf, 2020.

Vergos, G. S., Erol, B., Natsiopoulos, D. A., Grigoriadis, V. N., Isik, M. S., and Tziavos, I. N.: Preliminary Results of GOCE-Based Height System Unification between Greece and Turkey over Marine and Land Areas, Acta Geodaetica Et Geophysica, 53, 61–79, https://doi.org/10.1007/s40328-017-0204-x, wOS:000429387700005, 2018.

Wahr, J., Molenaar, M., and Bryan, F.: Time variability of the Earth's gravity field: Hydrological and oceanic effects and their possible detection using GRACE, Journal of Geophysical Research: Solid Earth, 103, 30 205–30 229, https://doi.org/10.1029/98JB02844, http://doi.wiley.com/10.1029/98JB02844, 1998.

Yi, W.: An Alternative Computation of a Gravity Field Model from GOCE, Advances in Space Research, 50, 371 – 384, https://doi.org/10.1016/j.asr.2012.04.018, 2012.

Yi, W., Rummel, R., and Gruber, T.: Gravity Field Contribution Analysis of GOCE Gravitational Gradient Components, Studia Geophysica et Geodaetica, 57, 174–202, https://doi.org/10.1007/s11200-011-1178-8, 2013.

785 Zehentner, N. and Mayer-Gürr, T.: Precise Orbit Determination Based on Raw GPS Measurements, Journal of Geodesy, 90, 275–286, https://doi.org/10.1007/s00190-015-0872-7, 2016.

Zehentner, N. and Mayer-Gürr, T.: Precise orbit determination based on raw GPS measurements, Journal of Geodesy, 90, 275–286, https://doi.org/10.1007/s00190-015-0872-7, 2016.

Zingerle, P., Pail, R., Gruber, T., and Oikonomidou, X.: The Experimental Gravity Field Model XGM2019e,
790 https://doi.org/10.5880/ICGEM.2019.007, http://dataservices.gfz-potsdam.de/icgem/showshort.php?id=escidoc:4529896, 2019.

Zingerle, P., Pail, R., Gruber, T., and Oikonomidou, X.: The Combined Global Gravity Field Model XGM2019e, Journal of Geodesy, https://doi.org/10.1007/s00190-020-01398-0, 2020.